# Learning Unmasking Policies for Diffusion Language Models

Metod Jazbec [* 1]  Theo X. Olausson [* 2]  Louis Béthune [3]  Pierre Ablin [3]  Michael Kirchhof [3]
João Monteiro [3]  Victor Turrisi [3]  Jason Ramapuram [3]  Marco Cuturi [3]

## Abstract

Diffusion (Large) Language Models (dLLMs) now match the downstream performance of their autoregressive counterparts on many tasks, while holding the promise of being more efficient during inference. One critical design aspect of dLLMs is the *sampling procedure* that selects which tokens to unmask at each diffusion step. Indeed, recent work has found that heuristic strategies such as confidence thresholding improve both sample quality and token throughput compared to random unmasking. However, such heuristics have downsides: they require manual tuning, and we observe that their performance degrades with larger block sizes. In this work, we instead propose to train sampling procedures using reinforcement learning. Specifically, we formalize masked diffusion sampling as a Markov decision process in which the dLLM serves as the environment, and propose a lightweight policy based on a single-layer transformer that maps dLLM token confidences to unmasking decisions. Our experiments show that these trained policies match the performance of state-of-the-art heuristics when combined with semi-autoregressive (block) generation, while outperforming them in the full-diffusion setting. Our code is available at https://github.com/apple/ml-rl-dllm.

## 1. Introduction

Discrete diffusion (Austin et al., 2021a; Hoogeboom et al., 2021; Dieleman et al., 2022; Lou et al., 2024; Shi et al., 2024) has recently emerged as a compelling alternative to the predominant autoregressive (AR) modeling paradigm in large language models (LLMs). Unlike AR models, which

---

* Equal contribution; work done during internships at Apple. [1]University of Amsterdam [2]Massachusetts Institute of Technology [3]Apple. Correspondence to: <m.jazbec@uva.nl>, <theoxo@csail.mit.edu>, <cuturi@apple.com>.

*Proceedings of the 43rd International Conference on Machine Learning*, Seoul, South Korea. PMLR 306, 2026. Copyright 2026 by the author(s).

generate the next token in a left-to-right fashion (Radford et al., 2018), diffusion LLMs (dLLMs) generate text by learning to reverse a noising process that progressively corrupts token sequences. In particular, masked diffusion models (MDMs; Sahoo et al. 2024; Shi et al. 2024; Ou et al. 2025), a subclass of discrete diffusion models that use time-dependent BERT-style (Devlin et al., 2019) masking as the forward noising process, have recently demonstrated impressive performance, with models like LLaDA (Nie et al., 2025) and Dream (Ye et al., 2025) matching the performance of similarly sized autoregressive LLMs.

At generation time, MDMs begin with a fully masked sequence and iteratively unmask a fixed number of tokens at randomly sampled positions in each sampling step. As a result, they offer the potential for faster inference, as they can, in principle, generate multiple tokens in parallel using a single model call. Despite this promise, open-source dLLMs have, until recently, lagged behind their AR counterparts in terms of inference efficiency. This changed with Fast-dLLM (Wu et al., 2025), which demonstrated that a dLLM like LLaDA (Nie et al., 2025) can achieve higher token throughput than similarly sized LLaMA models (Dubey et al., 2024) while maintaining competitive performance. A key component of Fast-dLLM lies in its sampling heuristic: instead of unmasking a fixed number of randomly sampled token positions, it proposes to unmask all tokens whose confidences exceed a pre-specified threshold. This success has since inspired the development of increasingly sophisticated sampling heuristics (see Appendix A for a comprehensive overview), further advancing the state of the art. These heuristics can, however, be difficult to tune, and seem to work best when imposing semi-autoregressive (semi-AR) generation, in which small blocks of tokens are unmasked sequentially (Arriola et al., 2025a; Nie et al., 2025). We posit that such limitations are difficult to avoid with heuristics alone, since they are essentially handcrafted solutions to a sequential decision making problem: *What is the optimal order in which to unmask the sequence of tokens, in order to strike a good balance between correctnes and efficiency?*

We address this question by moving beyond heuristics, instead proposing a learning-based approach that leverages a transformer-based unmasking policy trained via reinforcement learning (RL). This approach is motivated by the above

observation that unmasking in dLLMs can be viewed as a (Markovian) sequential decision making problem, and follows recent successes in RL for post-training of language models. However, in contrast to recent works that use RL to improve the reasoning abilities of dLLMs (e.g., Zhao et al. 2025b), our goal is not to improve the capacity of the underlying model, but rather to use RL to automate the discovery of adaptive sampling strategies. We therefore treat the underlying dLLM as the *environment* in which to act, not the policy, leaving it unchanged and instead parameterizing a policy as a lightweight stand-alone network (Figure 2). Empirically, we demonstrate that our learned sampling policies match the performance of state-of-the-art heuristic samplers (Wu et al., 2025) in standard semi-AR generation, while also surpassing heuristics in full-diffusion setting. In summary, we make the following contributions:

- We formalize sampling in dLLMs as a Markov decision process (MDP) (Section 3.1), propose a lightweight design for the unmasking policy (Section 3.2), and outline our RL training pipeline based on group relative policy optimization (GRPO; Shao et al. 2024) (Section 3.3).
- Our experiments show that learned policies with our RL framework match the performance of heuristic samplers such as Fast-dLLM (Section 4.1), while also addressing some of the challenges heuristic samplers face outside the semi-AR regime (Section 4.2).
- We study the transferability of our learned samplers across different models and data domains (Section 4.3), and provide detailed insights into stabilizing RL training and policy design (Section 4.4). We also demonstrate that policies exhibit qualitatively different unmasking behavior compared to confidence heuristics (Appendix D).

## 2. Background

Throughout the paper, we use the notation $[L]$ to represent the set $\{1, \ldots, L\}$. We denote a sequence of tokens as $\boldsymbol{x} = (x^1, \ldots, x^L) \in \mathcal{V}^L$, where $L$ is the sequence length and $\mathcal{V} := [V]$ is the vocabulary. Given our focus on MDMs, we assume the vocabulary includes a special mask token $\boldsymbol{M}$.

### 2.1. Masked Diffusion Models

We focus on masked diffusion models (MDMs), deferring a more general introduction to discrete diffusion to Appendix H.

**Training.** MDMs learn to generate data by training a BERT-style (Devlin et al., 2019) masked predictor to reverse the forward noising process. Concretely, given a training sample $\boldsymbol{x}_0 \sim p_{\text{data}}$, the forward process corrupts each token independently by setting it to the mask token with probability proportional to the diffusion timestep $t \in [0, 1]$:

$$p_t(\boldsymbol{x}_t \mid \boldsymbol{x}_0) := \prod_{k=1}^{L} p_t(x_t^k \mid \boldsymbol{x}_0),$$

$$\text{where } p_t(x_t^k \mid \boldsymbol{x}_0) := \begin{cases} 1 - t, & \text{if } x_t^k = x_0^k, \\ t, & \text{if } x_t^k = \boldsymbol{M}, \\ 0, & \text{otherwise.} \end{cases}$$

Recent work has shown that an MDM parametrized by $\theta$ can learn the reverse process by maximizing the following evidence lower bound (ELBO) (Sahoo et al., 2024; Shi et al., 2024; Ou et al., 2025), $\mathcal{L}(\theta) \leq \mathbb{E}_{\boldsymbol{x}_0 \sim p_{\text{data}}}[\log p_\theta(\boldsymbol{x}_0)]$:

$$\mathcal{L}(\theta) := \mathbb{E}_{t \sim U[0,1], \, \boldsymbol{x}_0 \sim p_{\text{data}}, \, \boldsymbol{x}_t \sim p_t(\boldsymbol{x}_t \mid \boldsymbol{x}_0)}$$

$$\left[ \frac{1}{t} \sum_{k=1}^{L} \mathbf{1}[x_t^k = \boldsymbol{M}] \log p_\theta(x_0^k \mid \boldsymbol{x}_t) \right] \quad (1)$$

with $\mathbf{1}[\cdot]$ denoting the indicator function.

**Generation.** When generating an answer for a given prompt $\boldsymbol{x}$, MDMs start with a sequence of all-masked tokens $\boldsymbol{y}_T := (\boldsymbol{M}, \ldots, \boldsymbol{M})$, where $L := |\boldsymbol{y}_T|$ is a pre-specified maximum answer length. Then, in each sampling step $t \in [T]$, the MDM outputs token distributions $p_t^k := p_\theta(y_0^k = \cdot \mid \boldsymbol{x}, \boldsymbol{y}_t)$ for all token positions $k \in [L]$ and a sampling strategy decides which subset $\mathcal{U}_t \subseteq \mathcal{M}_t$ of the still-masked positions $\mathcal{M}_t := \{k \in [L] \mid y_t^k = \boldsymbol{M}\}$ to unmask. A new, partially unmasked/denoised answer $\boldsymbol{y}_{t-1}$ is obtained via

$$y_{t-1}^k := \begin{cases} y \sim p_t^k, & \text{if } k \in \mathcal{U}_t, \\ y_t^k, & \text{otherwise.} \end{cases} \quad (2)$$

The stochasticity of sampling $y \sim p_t^k$ depends on the dLLM temperature $\tau$, with higher temperatures leading to more diverse (but often lower quality) generations. When evaluating dLLMs, it is common to set $\tau = 0$ (Nie et al., 2025; Wu et al., 2025), resembling greedy decoding. The choice of sampling/unmasking strategy is also important, since different choices will produce different unmasking sets $\mathcal{U}_t$, thus affecting both sampling quality and efficiency. This has drawn considerable attention to the development of optimal sampling techniques for dLLMs (see Appendix A).

### 2.2. Heuristic Samplers

One popular approach to decide which positions $\mathcal{U}_t$ to unmask at each timestep is to construct a heuristic that leverages uncertainty measures derived from the predicted token distributions $p_t^k$ (Kim et al. 2025a; Wu et al. 2025; Ben-Hamu et al. 2025; *inter alia*). In particular, recent work has shown substantial efficiency gains with heuristic strategies that employ the *confidence* $c_t^k := \max_{v \in \mathcal{V}} p_t^k(v)$ of the underlying dLLM in order to make their decisions. Two representative methods within this space are *high-confidence* unmasking (Chang et al., 2022), in which each forward pass

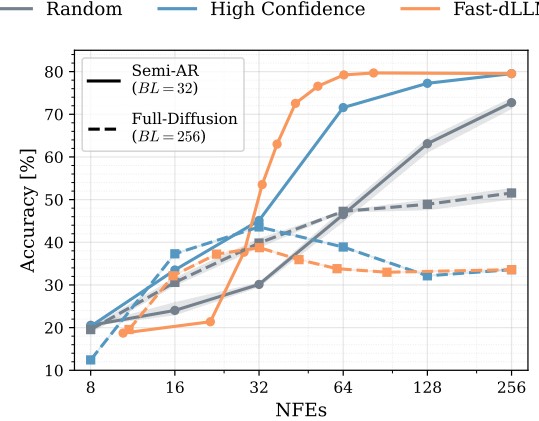

*Figure 1.* LLaDA-8B-Instruct (Nie et al., 2025) on GSM8k, with semi-AR generation ($BL = 32$; **—**) and without (full-diffusion regime, $BL = L = 256$; **- -**). More datasets and models in Appendix C.1. Generation speed is measured in network function evaluations (NFEs), which corresponds to the number of sampling steps.

unmasks a fixed number of tokens ($K$) with the highest confidences,

$$\mathcal{U}_t^K := \underset{I \subseteq \mathcal{M}_t, \, |I|=K}{\arg\max} \sum_{k \in I} c_t^k$$

as well as the confidence-thresholding strategy of Fast-dLLM (Wu et al., 2025), which allows for a variable number of tokens to be unmasked at each step by comparing the confidences to a fixed threshold $\lambda$:

$$\mathcal{U}_t^\lambda := \{k \in \mathcal{M}_t \mid c_t^k > \lambda\}.$$

Despite the undeniable success of confidence-based samplers, relying on handcrafted solutions comes with drawbacks. Beyond the obvious need for careful design (e.g., selecting an appropriate confidence measure or threshold $\lambda$), we also found that these methods often exhibit high sensitivity to the exact sampling configuration. For instance, many heuristics rely on semi-autoregressive (semi-AR) generation (Arriola et al., 2025a; Nie et al., 2025), where diffusion decoding proceeds one "block" of contiguous tokens at a time, with block length $BL$ serving as yet another hyperparameter. As shown in Figure 1, outside this semi-AR regime, the performance of confidence-based heuristics can degrade below that of a random unmasking, and no longer benefit from increased compute (i.e., higher numbers of function evaluations, NFEs). These limitations raise the question of whether effective sampling methods can be *learned* rather than manually tuned.

# 3. Learning Unmasking Policies

We now introduce our approach to learn sampling in dLLMs via reinforcement learning (RL). We begin by formulating sampling as a Markov decision process (MDP) (Section 3.1), then propose a sampling policy (Section 3.2), and finally provide details on our RL training (Section 3.3).

## 3.1. dLLM Sampling as a Markov Decision Process

To facilitate the development of RL-based samplers, we describe first the Markov decision process (MDP) (Sutton et al., 1998) for sampling in dLLMs. To stay aligned with the MDM notation in Section 2.1, we reverse the time in our MDP formulation: $t = T, T - 1, \ldots, 1$, with $T$ denoting the maximum horizon (number of sampling steps).

- The *state* $(\boldsymbol{x}, \boldsymbol{y}_t)$ holds a prompt $\boldsymbol{x}$ and the current (partially masked) generation $\boldsymbol{y}_t$. For brevity, we omit $\boldsymbol{x}$ from the state notation unless needed. The initial state has a fully masked generation: $\boldsymbol{y}_T = (\boldsymbol{M}, \ldots, \boldsymbol{M})$.
- An *action* $\boldsymbol{u}_t \in \{0,1\}^L$ is a vector of unmasking decisions indicating which positions have been selected to be unmasked in the next transition step. These actions are chosen according to the policy $\pi_\phi$: $\boldsymbol{u}_t \sim \pi_\phi(\cdot \mid \boldsymbol{y}_t)$ (introduced later in Section 3.2).
- The *transition* $\boldsymbol{y}_{t-1} \sim P(\boldsymbol{y}_{t-1} \mid \boldsymbol{y}_t, \boldsymbol{u}_t; \tau)$ corresponds to standard sampling in dLLMs (cf. Equation (2)), with the action $\boldsymbol{u}_t$ determining which tokens get unmasked: $\mathcal{U}_t^\pi := \{k \in \mathcal{M}_t \mid u_t^k = 1\}$.
- The *reward* $R(\boldsymbol{y}, \boldsymbol{y}_t)$ is provided only at the final generation step, which corresponds to the first timestep where all tokens have been unmasked $\hat{T} := \max\{t \in [T] \mid y_t^k \neq \boldsymbol{M}, \forall k \in [L]\}$. To learn useful samplers, the reward should promote both correctness (i.e., the generated answer $\boldsymbol{y}_{\hat{T}}$ being 'close' to the reference answer $\boldsymbol{y}$) and efficiency (i.e., minimizing the number of steps $T - \hat{T}$). Note that the reward depends on action $\boldsymbol{u}_t$, and therefore on the policy $\pi_\phi$, implicitly through its influence on the generations $\boldsymbol{y}_t$ and the number of steps. We defer our concrete choice of the reward function to Section 3.3.

With the MDP above, finding the optimal sampling becomes a standard RL problem of finding a policy $\pi_\phi$ parametrized by $\phi$ that maximizes the expected reward:

$$\max_\phi \mathbb{E}_{(\boldsymbol{x}, \boldsymbol{y}) \sim p_{\text{data}}, \, \boldsymbol{u}_t \sim \pi_\phi, \, \boldsymbol{y}_t \sim P} \left[ \sum_{t=1}^{T} R(\boldsymbol{y}, \boldsymbol{y}_t) \right].$$

## 3.2. Lightweight Confidence Policy Design

Having outlined the MDP above, we next describe our proposed implementation for the sampling policy $\pi_\phi$ (see Figure 2 for policy sampling diagram and Appendix B for sampling algorithm).

**Confidence-based input.** Recall that a state consists of a partially masked token sequence $\boldsymbol{y}_t$. To avoid construct-

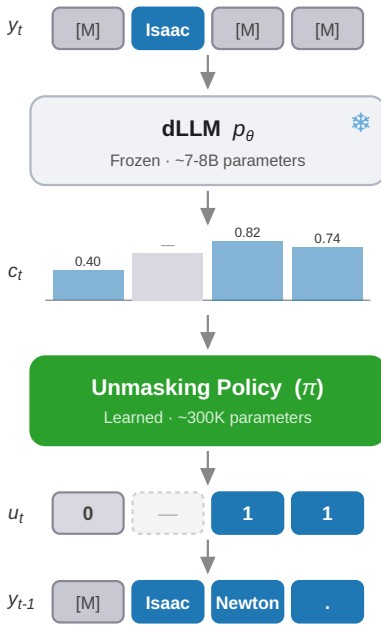

Figure 2. Diagram showing how our policies are used on top of a pretrained dLLM to unmask tokens and generate text.

ing policies that operate at the token level, and thereby minimize computational overhead, we rely on the vector of token confidences $c_t := (c_t^1, \ldots, c_t^L)$, which are readily available since the token-level predictive distributions $p_t^k$ are computed at each transition step $y_{t-1} \sim P$ anyways. Our choice is further motivated by the aforementioned heuristic methods that primarily operate on confidences (Wu et al., 2025), and also based on our own ablations (see Section 4.4). Specifically, we observed that alternatively relying on dLLM's hidden states required significantly larger policy models without consistently improving performance.

**Lightweight transformer.** We introduce a small, learnable neural network $f_\phi$ that maps the vector of confidences $c_t$ to a vector of unmasking scores (logits) $b_t = f_\phi(c_t, m_t, t)$ with $b_t \in \mathbb{R}^L$. We additionally include a binary mask vector $m_t := (m_t^1, \ldots, m_t^L)$, where $m_t^k := \mathbf{1}[k \in \mathcal{M}_t]$ indicates whether position $k$ is still masked, and inform the policy with the time index $t \in [T]$. In practice, $f_\phi$ is implemented as a lightweight transformer (see Section 4 for more details on the exact policy architecture). Notably, its size is less than $0.01\%$ of the pretrained dLLMs used in our experiments, resulting in negligible computational overhead during sampling (see Figure 12).

**Bernoulli likelihood.** We sample the unmasking actions via $u_t^k \sim \mathrm{Ber}(s_t^k)$ with $s_t^k := \sigma(b_t^k)$, where $\mathrm{Ber}(\cdot)$ denotes the Bernoulli distribution and $\sigma(\cdot)$ is the sigmoid function. Conveniently, the policy likelihood $\pi_\phi(u_t) := \prod_{k=1}^L (s_t^k)^{u_t^k} \cdot (1 - s_t^k)^{1-u_t^k}$ is readily available in closed

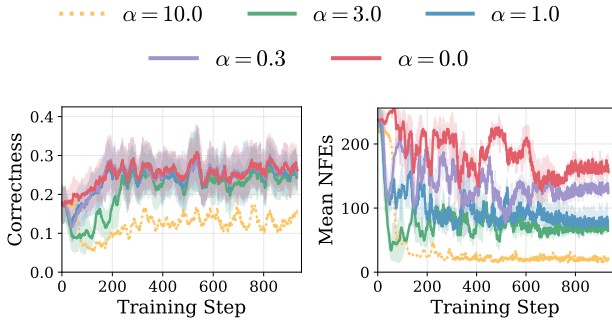

Figure 3. Correctness reward (rolling average, 20 steps) on GSM8k (*left*) and average number of sampling steps (*right*) during training of our policies for various values of $\alpha$ (cf. Equation (3)). Averaged over two random seeds, with shaded areas indicating (min, max); only one seed shown for $\alpha = 10.0$ (⋯) due to training instability.

form and does not require any additional approximations (unlike in the case of post-training dLLMs; Zhao et al. 2025b). We also considered an alternative formulation based on the Plackett-Luce model (Luce, 1959; Plackett, 1975) which we call *dynamic Plackett-Luce sampling* (DPLS; see Appendix F), but since our ablations showed comparable performance (Section 4.4), we favor the Bernoulli formulation in our experiments for its simplicity.

At generation, we additionally propose to "temper" the Bernoulli probabilities as $s_t^k := \sigma(b_t^k / \tau_\pi)$. The temperature $\tau_\pi$ thus controls the sharpness of the policy distribution; we found that this can sometimes serve as a useful test-time knob for trading off accuracy versus efficiency by making the policy more or less "decisive" (Section 4).

To ensure convergence when all sampled actions are zero ($u_t = 0$), we unmask the position with the highest Bernoulli parameter $s_t^k$ (similar to Fast-dLLM). We apply this fallback only at test time, as we found that forcing unmasking during training was prone to reward hacking. Instead, if not all tokens are unmasked within $T$ steps during training, we simply terminate generation and evaluate the answer with some tokens left unmasked. Since such samples tend to get low or no reward, we find that the policies quickly learn not to leave any tokens unmasked without explicit instruction.

### 3.3. Learning $\pi_\phi$ with GRPO

To train our sampling policy, we adopt group relative policy optimization (GRPO) (Shao et al., 2024), a method recently popularized as a simpler and more scalable alternative to earlier policy gradient approaches such as PPO (Schulman et al., 2017). Specifically, for each prompt $x \in \mathcal{D}$, we sample $G$ trajectories of generations $\{y_T^g, \ldots, y_{\hat{T}_g}^g\}_{g=1}^G$ along with their corresponding unmasking decisions $\{u_T^g, \ldots, u_{\hat{T}_g}^g\}_{g=1}^G$. Importantly, we fix the dLLM sampling temperature $\tau$ to 0 (i.e., greedy decoding)

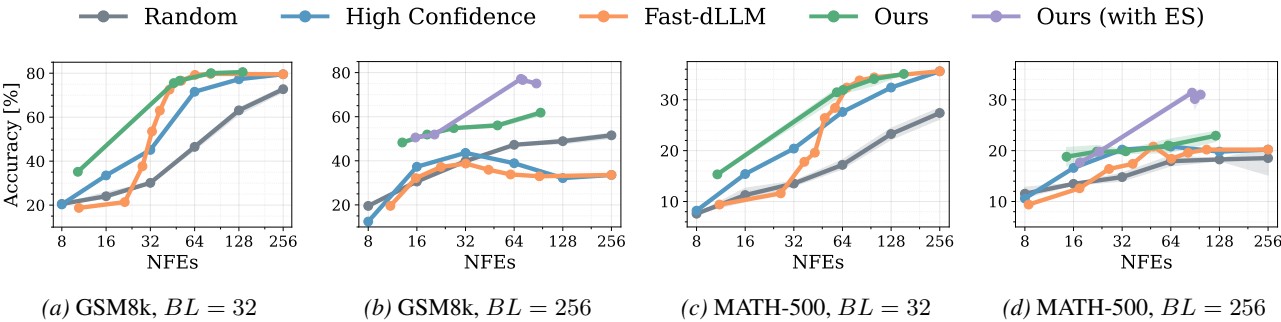

*Figure 4.* Results for LLaDA in semi-AR (Figure 4a & Figure 4c) and full-diffusion (Figure 4b & Figure 4d) generation regimes ($L = 256$). Results for Dream-7B are provided in Figure 13. For our policies we vary $\alpha \in \{10, 3, 1, 0.3, 0\}$ and use $\tau_\pi = 0.5$ for $BL = 32$ and $\tau_\pi = 1$ for $BL = 256$. Expert steering (ES) described in detail in Appendix G. Wall-clock time plots shown in Figure 12. For plot readability, we leave out some baselines (Kim et al., 2025a; Ben-Hamu et al., 2025) here and provide the comparison with them in Figure 11.

to ensure that any variation among samples within a group arises solely from different unmasking actions. After computing rewards for each sample in the group, we define the advantage as $A_t^g := R(\boldsymbol{y}, \boldsymbol{y}_{\hat{T}_g}^g) - \frac{1}{G} \sum_{i=1}^{G} R(\boldsymbol{y}, \boldsymbol{y}_{\hat{T}_i}^i)$, following recent best practices that recommend omitting standard deviation normalization of advantages (Zhao et al., 2025b). Additionally, note that the reward at the final generation step $\hat{T}_g$ for each sample is propagated to all preceding timesteps $t$ to provide a learning signal throughout the entire sampling process. Our final training objective is then

$$\mathcal{J}(\phi) := \mathbb{E}_{(\boldsymbol{x}, \boldsymbol{y}) \sim \mathcal{D}, \{\boldsymbol{y}_{T:\hat{T}_g}^g\}_{g=1}^{G} \sim P_\theta, \{\boldsymbol{u}_{T:\hat{T}_g}^g\}_{g=1}^{G} \sim \pi_{\phi_{\text{old}}}}$$

$$\left[ \frac{1}{G} \sum_{g=1}^{G} \frac{1}{T - \hat{T}_g} \sum_{t=\hat{T}_g}^{T} \min \left\{ \rho_t^g \cdot A_t^g, \right. \right.$$

$$\left. \left. \text{clip}(\rho_t^g, 1 - \epsilon, 1 + \epsilon) \cdot A_t^g \right\} \right]$$

where $\pi_\phi$ is the current policy being updated, and $\pi_{\phi_{\text{old}}}$ refers to an earlier version of the policy used to generate the RL rollouts. The likelihood ratio $\rho_t^g := \frac{\pi_\phi(\boldsymbol{u}_t^g)}{\pi_{\phi_{\text{old}}}(\boldsymbol{u}_t^g)}$ serves as the importance sampling correction term, which together with clipping (via $\epsilon$) aims to stabilize the off-policy training. Note that we exclude already-unmasked positions from the policy likelihood computation. Finally, since our policies are trained from scratch, we remove the KL regularization term in our GRPO objective.

**Reward.** As mentioned in Section 3.1, we wish to obtain a policy that yields 'correct' generations while also being fast. To this end, we define our final *multiplicative* reward as

$$R(\boldsymbol{y}, \boldsymbol{y}_t) := \begin{cases} r(\boldsymbol{y}, \boldsymbol{y}_t) \cdot \left(1 - \frac{T-t}{T}\right)^\alpha, & \text{if } t = \hat{T}, \\ 0, & \text{otherwise,} \end{cases} \quad (3)$$

where $r(\boldsymbol{y}, \boldsymbol{y}_t)$ is a task-specific correctness term (e.g., a binary reward indicating whether the generated mathemat-

ical answer is correct). To encourage faster sampling, we incorporate a computational penalty based on the number of steps, $T - \hat{T}$, with $\alpha \geq 0$ serving as a hyperparameter that controls the trade-off between accuracy and speed. While we experimented with an additive penalty of the form $r(\boldsymbol{y}, \boldsymbol{y}_{\hat{T}}) - \alpha \left(\frac{T-\hat{T}}{T}\right)$ (Graves, 2016) we found that this led to problematic reward hacking: when all samples in a group are incorrect, as is the case early on when training from scratch, faster samples may still receive a positive advantage, despite being wrong (cf. Section 4.4).

## 4. Experiments

We begin by verifying that the learned policies match the performance of confidence-based heuristics (Section 4.1). We then highlight the potential of RL sampling to outperform heuristics in the full-diffusion setting (Section 4.2), and examine the transferability of RL policies across models, datasets, and generation lengths (Section 4.3). Next, we explore different ways to instantiate the MDP and analyze their effects on performance (Section 4.4). Finally, we study the unmasking behavior of RL policies, highlighting their qualitative differences to heuristics (details in Appendix D).

### 4.1. Learning Effective Sampling via RL

We start by demonstrating that our proposed RL framework yields effective sampling strategies in dLLMs, where effectiveness is defined in terms of both accuracy and speed.

**Experiment details.** We use LLaDA-8B-Instruct (Nie et al., 2025) as the base dLLM (additional results on Dream-7B-Instruct (Ye et al., 2025) provided in Appendix C.4). We parametrize the policy network $f_\phi$ as a shallow (single-layer) transformer incorporating adaptive layer normalization for conditioning; hyperparameter details for the architecture are provided in Appendix I and an architecture diagram can be found in Appendix J. We train five differ-

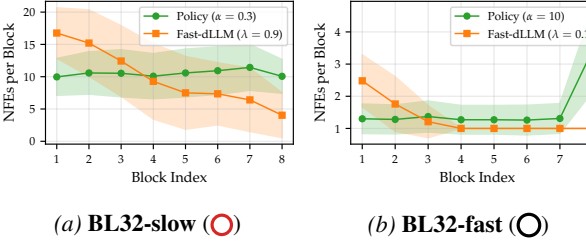

*(a)* **BL32-slow** (⃝)       *(b)* **BL32-fast** (⃝)

*Figure 5.* Average number of sampling steps (NFEs) per block for LLaDA on GSM8K with semi-AR generation (same setting as in Figure 4a). *Left* we show NFEs per block for 'slow' variants of Fast-dLLM and policy sampling ($\sim 75$ NFEs), while in the *right* plot we show 'fast' variants ($\sim 10$ NFEs). Fast-dLLM exhibits a pattern of allocating more compute to earlier blocks. Our policy sampling, by contrast, distributes compute more uniformly across blocks, except in the fast ($\alpha = 10$) policy, where most compute is expended in the final block while generating numerical answers. To see where evaluated policies here are located on the pareto frontier, see Figure 21a. More details on qualitative differences between Fast-dLLM and policy sampling are provided in Appendix D.

ent policies, corresponding to $\alpha \in \{10, 3, 1, 0.3, 0\}$. Each policy is trained semi-autoregressively at $BL = 32$ on a single epoch of mixture data, sampled proportionally from the training sets of GSM8K (Cobbe et al., 2021) and MATH (Hendrycks et al., 2021), resulting in roughly 15,000 training samples. Figure 3 shows the training dynamics; as expected, higher $\alpha$ generally gives rise to faster policies.

We then compare the resulting policies on the test sets of both GSM8K and MATH to the confidence-based heuristics introduced in Section 2.2, picking one out of two training seeds based on the final training loss and averaging over three test-time seeds. To obtain Pareto frontiers for the heuristic methods, we use $K \in \{8, 16, 32, 64, 128, 256\}$ for the random baseline and high-confidence unmasking, while for Fast-dLLM we use $\lambda \in \{0.1, 0.2, \ldots, 1.0\}$ throughout. For all methods, we use the standard greedy decoding setting ($\tau = 0$) when generating test answers.

**Results for (Short)** $BL = 32$, **Figure 4a and Figure 4c.** Generally, we observe that the performance of the learned policies (—) exceeds those of both the random baseline (—) as well as the high-confidence sampling (—), while matching Fast-dLLM (—). That our learned policies do not surpass Fast-dLLM in the mid-to-high NFEs range may suggest that this heuristic is near-optimal under semi-AR generation. Interestingly, despite very similar quantitative performance, we find that the learned policies exhibit qualitatively somewhat different unmasking behavior compared to Fast-dLLM (see Appendix D). Concretely, the policy unmasks adjacent tokens much less frequently (Figure 24) and allocates compute more uniformly across blocks (Figure 5a).

We note that the effect of $\alpha$ in training (cf. Figure 3) carries over to the test set: policies trained with higher values of $\alpha$ exhibit faster, yet less accurate, sampling. Of particular note

is $\alpha = 10.0$, which exhibited greater training instability due to the very sharp slope of the reward. When successfully trained, this yields a very fast policy that outperforms Fast-dLLM in the low-NFE ($\sim 10$) regime on both datasets. Qualitatively, we find evidence that this improved performance is likely due to the policy learning to slow down in the final block when generating a numerical answer, as shown in Figure 5b (see also Appendix D for more details). This highlights the potential of RL-based policies when optimizing for maximal efficiency.

However, RL policies appear to exhibit less *controllability*, as varying $\alpha$ (at train-time) results in a less smooth traversal of the Pareto frontier than varying the confidence threshold $\lambda$ in Fast-dLLM (at test-time). For example, the behavior of the $\alpha = 3.0$ policy is nearly identical to that of $\alpha = 1.0$, despite the reward incorporating a computational penalty that scales exponentially with $\alpha$. Furthermore, we find that this cannot easily be remedied by using a tighter $\alpha$ grid. In Appendix C.5 we let $\alpha$ range in $\{10.0, 9.0, \ldots 1.0, 0.3, 0.0\}$, but observe that using $\alpha \geq 4.0$ consistently results in converging to a policy either roughly equivalent to that of $\alpha = 3.0$ or to that of $\alpha = 10.0$, with no value of $\alpha$ successfully yielding a policy which interpolates between the two extremes.

To test whether the speed-accuracy trade-off can be controlled at test-time instead, we experiment with post-hoc scaling of the unmasking probabilities obtained from a policy trained with $\alpha = 1$. Concretely, following Chen et al. (2025a) we introduce a free parameter $\beta > 0$ and scale the Bernoulli probabilities $s_t^k$ as $u_t^k \sim \text{Ber}(\min\{1, \beta \cdot s_t^k\})$. When $\beta > 1$, this increases the average number of tokens unmasked per step, whereas $\beta < 1$ decreases it. By varying $\beta$, the behavior of a policy trained at a fixed $\alpha$ can thus be modulated without retraining. Encouragingly, as shown in Figure 6, this turns out to be an effective test-time knob, smoothly navigating the accuracy-efficiency frontier. At mid-to-high NFEs (low $\beta$) the policy mostly matches the Fast-dLLM semi-AR frontier, while outperforming it at lower NFEs (high $\beta$). For example, on MATH-500 at around 25 NFEs, the policy achieves  20% accuracy compared to Fast-dLLM's  10%. Besides $\beta$, we also observe that one can make small changes to the trade-off between compute and performance by varying the policy temperature $\tau_\pi$ at test time; we detail the impact of this parameter in Figure 16.

### 4.2. Beyond Semi-Autoregressive Decoding

As mentioned in Section 2.2, heuristic samplers often rely on semi-AR generation to achieve good performance. While not typically thought of as problematic in textual domains where strong AR dependencies exist, semi-AR approaches can only partially fulfill the promise of fully parallel generation by restricting the unmasking to the current block. Hence, we train here policies without semi-AR generation, targeting the full-diffusion task ($BL = L = 256$) at both

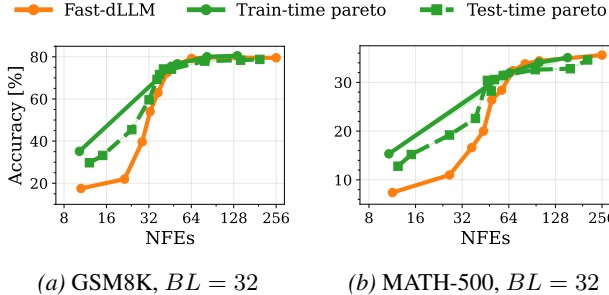

*Figure 6.* Scaling unmasking probabilites at test-time as $u_t^k \sim$ $\mathrm{Ber}(\min\{1, \beta \cdot s_t^k\})$ for $\beta \in \mathbb{R}_+$ leads to smoother pareto froniter compared to changing $\alpha$ (Equation (3)) at train-time. Results for LLaDA in semi-AR regime. Note that results for Fast-dLLM and train-time pareto of policy sampling are reproduced from Figure 4.

training and test time. We keep all other implementation details identical to those in the previous section.

**Results for (Long)** $BL = 256$, **Figure 4b & Figure 4d.** Although all sampling strategies experience a performance drop relative to the semi-AR setting (cf. Figure 4a & Figure 4c), we find that the policies produced by RL (—) exhibit the smallest decline and consequently achieve the best overall performance. Furthermore, the results suggest that this methodology is able to produce policies which achieve solid performance even in the low-NFE regime. In particular, the learned policies obtain $\sim 50\%$ accuracy at $\sim 12$ NFEs on GSM8K (compared to $\leq 30\%$ for the heuristic methods regardless of semi-AR use), highlighting the potential of non-semi-AR sampling for achieving maximal efficiency gains. Note however that, in theory, an RL policy trained with $BL = L$ could learn to emulate semi-AR sampling on its own. Thus, the fact that policies trained with $BL = L$ underperform those trained with $BL = 32$ in the mid-high NFE range (comparing, for example, Figure 4a to Figure 4b) suggests that the policies remain far from optimal.

One hypothesis for why this is the case is that our training procedure is not encouraging sufficient exploration, instead converging to locally optimal policies. To address this, we encourage further exploration by using samples generated via semi-AR sampling from Fast-dLLM. We refer to this approach as *expert steering*, and describe it in more detail in Appendix G. As shown in Figure 4, with expert steering (—) RL is able to discover policies that almost close the performance gap to the best accuracy achieved in the semi-AR setting at mid-to-high NFEs (e.g., $\sim 80\%$ on GSM8K and $\sim 35\%$ on MATH), while mostly retaining their strong performance in the low-NFE regime. However, we do find that expert steering introduces significant instability during training, further reducing the controllability through $\alpha$, with multiple values of $\alpha$ collapsing to near-identical policies. We leave further investigations into stabilizing expert steering training for future work. Finally, we observe that policy sampling and Fast-dLLM induce markedly different

unmasking orders in the full-diffusion setting (see Figure 7 and Appendix D). While Fast-dLLM exhibits reverse (right-to-left) unmasking, the policy with expert steering learns to overcome this behavior and unmasks earlier positions first on average (left-to-right), which likely explains its superior performance.

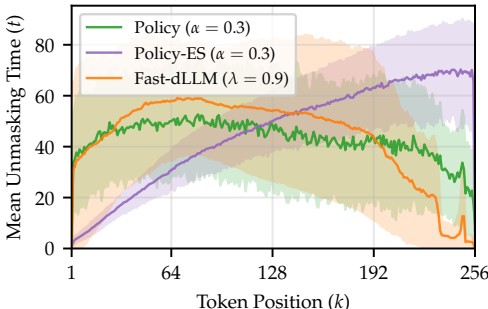

*Figure 7.* Mean unmasking time for each token position ($L = 256$), averaged over $N = 100$ samples, for LLaDA on GSM8K under full-diffusion generation (same setting as in Figure 4b). For visualization purposes, time is shown in reverse. Fast-dLLM (—) exhibits somewhat reverse (i.e., right-to-left) generation due to LLaDA's corrupted confidence on padding tokens (see Appendix D). Encouragingly, the expert steering policy (—) learns to overcome this issue and recovers left-to-right generation (observe how tokens at earlier positions are generated first on average). The locations of the evaluated policies on the Pareto frontier are shown in Figure 21b (see ◯).

### 4.3. Transferability of RL Sampling Policies

We next turn to the question of transferability. A notable advantage of heuristic approaches is that they can be applied *post-hoc* to any model or dataset.[1] This naturally raises the question: to what extent can RL policies trained on a specific model and dataset be reused in other settings?

**Model transfer: LLaDA → Dream.** We begin by investigating the transferability of RL policies across models; note that such transfers are possible because our policies rely solely on token confidences and are therefore agnostic to token embeddings, which would not transfer from one model to another. We thus reuse the policies trained on LLaDA (cf. Section 4.1) and evaluate them on Dream; the results are shown in Figure 8a (see Figure 17 for MATH). Encouragingly, most of the LLaDA-trained policies nearly match the performance of Fast-dLLM when evaluated on Dream, and perform very similarly to those trained on Dream directly (Figure 13). The one exception is the $\alpha = 10$ policy (✖), which collapses to Fast-dLLM performance and fails to retain the good low-NFE performance observed with LLaDA (cf. Figure 4a)—suggesting that the extreme steepness of

---

[1]Though heuristics might still require manual hyperparameter tuning, as exemplified by different optimal threshold across datasets for Fast-dLLM (e.g., comparing Figure 4a vs Figure 8b).

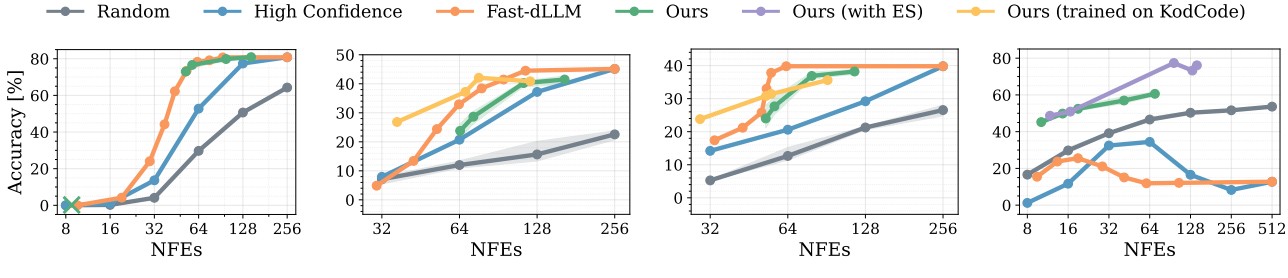

*(a)* Model transfer: LLaDA $\rightarrow$ Dream on GSM8K.

*(b)* Domain transfer: math $\rightarrow$ coding (HumanEval).

*(c)* Domain transfer: math $\rightarrow$ coding (MBPP).

*(d)* Sequence length transfer: $256 \rightarrow 512$ (GSM8K).

*Figure 8.* Results for the transferability experiments. Note that in **(a)**, the $\alpha = 10$ policy is represented separately by a (✕) in the lower left to avoid misleading visualization when interpolating to $\alpha = 3$. For results on coding datasets in **(b)** and **(c)**, we omit the low-NFE regime, as all approaches degrade to near-zero performance in this setting.

the reward curve (due to high $\alpha$) causes this policy to overfit to model-specific patterns in LLaDA's confidence levels.

**Domain transfer: Mathematical reasoning to code.** Next, we examine how well our policies transfer across different domains. We again reuse the policies from Section 4.1, which were trained on a mixture of mathematical data (GSM8K/MATH), but this time evaluate them on the coding tasks of HumanEval (Chen et al., 2021) and MBPP (Austin et al., 2021b). As shown in Figure 8b and Figure 8c (—), we find that these policies fail to fully transfer to the coding domains, especially on HumanEval. To investigate whether this drop in performance is due to the lack of domain-relevant training data, we train a new policy on the coding dataset KodCode-RL-10K (Xu et al., 2025) (—). These coding policies narrow the gap to the Fast-dLLM baseline on HumanEval and improve low-NFE performance on MBPP, underscoring the importance of using a diverse data mixture to support generalization across domains.

**Sequence length generalization:** $L = 256 \rightarrow L = 512$. We next study whether our policies transfer across different sequence lengths $L$ (not to be confused with the block length $BL$ in semi-AR). Such transfer is possible since we instantiate $f_\theta$ with a transformer, rather than, for example, an MLP (which requires fixed-length inputs). We thus take the policies from Section 4.2 and evaluate them at a 2x longer sequence length ($L = 512$). Results are shown in Figure 8d (see Figure 18 for MATH). While the baselines degrade further as the sequence length increases, the learned policies yield similar performance to before, suggesting that RL policies can transfer effectively across generation lengths without retraining.

**Non-greedy decoding ($\tau > 0$).** Lastly, we consider non-greedy decoding, i.e., using $\tau > 0$ when sampling the tokens in Equation (2).[2] To measure performance, we report

---

[2]In the non-greedy case, we define the token confidence as $c_t^k := p_\theta^k(y \mid \boldsymbol{x}, \boldsymbol{y}_t), \ y \sim p_\theta(\cdot \mid \boldsymbol{x}, \boldsymbol{y}_t; \tau)$.

pass@$k$ rates, which track the probability that at least one of the $k$ generations is correct. As shown in Figure 19, policy sampling yields higher pass@$k$ rates than Fast-dLLM sampling, and the gap widens with $k$ (+0.98% for $k = 1$ versus +2.56% for $k = 32$, on average across datasets). This shows that the policy's stochastic Bernoulli sampling explores the solution space more effectively and helps prevent the recently discovered diversity collapse of deterministic confidence-based samplers (Ni et al., 2026; Olausson et al., 2026). Beyond pass@$k$, policy sampling also dominates Fast-dLLM when selecting the final generation via self-consistency or an external reward model (Figure 19).

### 4.4. Exploring the Design Space of $\pi_\phi$

**Additive vs. multiplicative rewards.** We begin by ablating the structure of the reward function, comparing our proposed multiplicative combination of correctness and computational terms (cf. Equation (3)) with an additive alternative: $r(\boldsymbol{y}, \boldsymbol{y}_{\hat{T}}) - \alpha\left(\frac{T-\hat{T}}{T}\right)$. As shown in Figure 9a, while both exhibit increasing reward as training progresses, we observe that the additive formulation is much more prone to 'reward hacking', where it collapses to a very-fast-but-often-wrong policy. This is illustrated in Figure 9b: training with the additive reward results in a policy that unmasks everything at once, leading the number of sampling steps to collapse to the minimum possible for all inputs. As discussed in Section 3.3, we attribute this issue to the fact that incorrect but fast samples may still receive a positive advantage under the additive reward. In contrast, we find that the multiplicative reward effectively mitigates this issue by assigning a positive advantage only if the generation is correct, resulting in more stable and predictable training behavior.

**Policy likelihood.** Recall from Section 3.2 that we model the unmasking probability for each position using a Bernoulli distribution. This has the advantage of admitting very efficient inference and likelihood calculations, but relies on the network $f_\phi$ to implicitly embed relationships between different tokens in the scores $\boldsymbol{b}_t$, and runs the risk

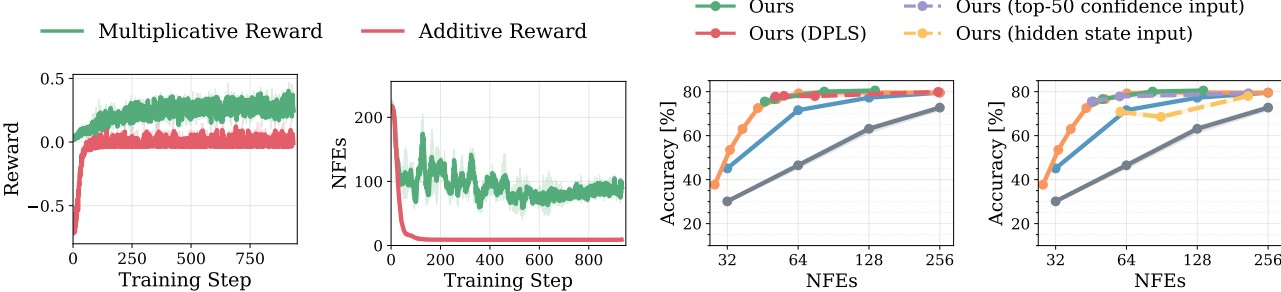

*(a)* Training reward for LLaDA with additive vs. multiplicative reward function (both $\alpha = 1.0$).  *(b)* Mean NFEs when training LLaDA with additive vs. multiplicative reward (both $\alpha = 1.0$).  *(c)* Bernoulli vs DPLS sampling. Both for LLaDA on GSM8K.  *(d)* Bernoulli policies with varying inputs. All for LLaDA on GSM8K.

*Figure 9.* Ablations for our proposed RL framework.

of producing $\mathcal{U}_t^\pi = \emptyset$ in case $\boldsymbol{u}_t = \boldsymbol{0}$. Here, we therefore investigate a more involved sampling procedure, which we call *dynamic Plackett-Luce sampling* (DPLS; detailed description in Appendix F) which is guaranteed to unmask at least one position in each step and which combines the scores $\boldsymbol{b}_t$ through a softmax. We retrain the policies from Section 4.1 using DPLS; the resulting downstream accuracy on GSM8K is then shown in Figure 9c. We observe that both methods achieve very similar performance, aligning closely with the Fast-dLLM, which might provide additional support for the hypothesis that this frontier is optimal in the semi-AR setting. Furthermore, DPLS policies appear to show slightly better controllability via $\alpha$ (as indicated by a larger spread of policies trained with varying $\alpha$).

**Confidence-based policy input.** Finally, we revisit our design choice of relying solely on the maximum token confidence values $c_t^k$ as input to the sampling policy (cf. Section 3.2). We first train and evaluate a set of policies which do not take only the highest confidence per position $c_t^k$ as input, but rather the top 50 highest values, thus giving the model more detailed information about the token predictive distributions $p_t^k$ and potentially allowing it to design its own confidences measures for effective sampling. The results are presented in Figure 9d. Somewhat surprisingly, using only the maximum confidence per position (—) performs slightly better than using the top-50 confidences (—), which suggests that alternative uncertainty measures are unlikely to yield performance gains over the simple confidences $c_t^k$.

Additionally, we explore parametrizing the policy as an additional classification head on top of LLaDA's final hidden state $\boldsymbol{h}_t^k \in \mathbb{R}^H$. While this results in a significantly larger policy ($300M$ parameters), it offers the potential advantage of incorporating token-level semantic information. However, in practice we observe that hidden state-based policies perform worse (see - - in Figure 9d) and exhibit less stable training dynamics than the confidence-based policies. These results suggest that the unembedding matrix $W \in \mathbb{R}^{H \times V}$,

which maps hidden states to token logits, plays a vital role in enabling effective policy decisions via confidence signals.

Lastly, we investigate whether additional inputs to the policy, such as the time step and mask vector, contribute meaningfully to performance. Concretely, we run a test-time experiment in which we zero out i) the time $t$, ii) the mask vector $\boldsymbol{m}_t$, or iii) both. As shown in Figure 20, the policy relies on both inputs: zeroing either leads to a drop in accuracy, with removing the mask causing the larger drop.

## 5. Conclusion

We introduced a reinforcement learning approach for learning unmasking strategies in diffusion LLMs. Our experiments demonstrated that the learned policies can match or even exceed the performance of recently proposed sampling heuristics, paving the way for the automated discovery of scalable and robust sampling mechanisms. Furthermore, we have given quantitative and qualitative insight into how to best train and deploy such unmasking policies across several reasoning and code domains.

**Limitations and future work.** Our core approach involves training a separate policy for each $\alpha$. Overall this is an effective strategy for trading off accuracy and efficiency, but it can be expensive and might not always yield sufficiently fine-grained control (Section 4.1). Our experiments suggest that test-time interventions may be a promising alternative (Figure 6 and Figure 16), but more work is needed to say for certain what the most effective and useful strategy for navigating the accuracy-efficiency frontier is. Beyond addressing these limitations, promising directions for future work include (i) expanding the training mixture to multiple domains; (ii) learning to remask previously unmasked tokens (Wang et al., 2025b; Huang et al., 2025c), and (iii) moving beyond text and applying our methodology to multimodal masked diffusion models (Swerdlow et al., 2025; Zhou et al., 2025; Bethune et al., 2026).

## Impact Statement

This paper presents work whose goal is to advance the field of Machine Learning. There are many potential societal consequences of our work, none which we feel must be specifically highlighted here.

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

# Appendix

The appendix is organized as follows:

- In Appendix A, we describe related work.

- In Appendix B, we provide an algorithm for our proposed policy sampling.

- In Appendix C, we present additional plots to supplement the experiments from the main paper:

    - In C.1, we replicate Figure 1 across models (LLaDA, Dream) and datasets (GSM8k, MATH).
    - In C.2, we replicate Figure 4 with more baselines.
    - In C.3, we replicate Figure 4 using latency (wall-clock time) as the efficiency measure.
    - In C.4, we replicate Figure 4 for policies trained on Dream.
    - In C.5, we show results for the same setting as Figure 4a and Figure 4c but with a denser $\alpha$-grid. We also show the spread of policies across different training seeds.
    - In C.6, we plot the impact of varying the policy temperature $\tau_\pi$.
    - In C.7, we provide model transfer results for the MATH dataset.
    - In C.8, we provide generation length (L) transfer results for the MATH dataset.
    - In C.9, we report results under non-greedy decoding.
    - In C.10, we conduct additional ablation experiments for our proposed policy design.

- In Appendix D, we perform a qualitative analysis of our learned sampling policies.

- In Appendix E, we visualize token generation trajectories.

- In Appendix F, we describe dynamic Plackett-Luce sampling as an alternative to Bernoulli sampling.

- In Appendix G, we detail our *expert steering* approach.

- In Appendix H, we discuss an extended background on discrete diffusion models.

- In Appendix I, we provide implementation and hyperparameters details.

- In Appendix J, we visualize the architecture of our sampling policy.

- Finally, in Appendix K, we include tabular versions of the main LLaDA results to simplify direct comparison in future work.

# A. Related work

**Heuristic Samplers for dLLMs.** Sampling in diffusion LLMs (dLLMs) (Nie et al., 2025; Ye et al., 2025) has recently attracted significant attention, with much of the work in this area proposing heuristic approaches to improve the decoding process in a training-free manner. Throughout this paper, we focus on Fast-dLLM (Wu et al., 2025) as a representative heuristic method, as it popularized confidence-thresholded sampling in dLLMs and demonstrated its crucial role in enabling faster inference in dLLMs compared to autoregressive models. Many recently proposed heuristics can be viewed as either reinterpretations or extensions of such confidence thresholding strategies (Ben-Hamu et al., 2025; Wei et al., 2025; Hong et al., 2025b; Li et al., 2025b; Yu et al., 2025; Kim et al., 2025b; Shen et al., 2025). Other notable heuristic approaches explore incorporating spatial (Huang et al., 2025a) or temporal information (Wang et al., 2025c), alternative confidence measures (Kim et al., 2025a), or more explicit modeling of token dependencies (Azangulov et al., 2025). In this work, we aim to complement ongoing research on sampling heuristics by investigating whether effective sampling strategies can be learned directly via reinforcement learning. Note that beyond improving the efficiency of unmasking in dLLMs, heuristics have also been proposed for remasking (Hong et al., 2025b; Dong et al., 2025) and for dynamically adjusting the generation length (Li et al., 2025a), which we do not target in this work.

**Reinforcement Learning Post-Training for dLLMs.** Early work on post-training diffusion LLMs via reinforcement learning includes d1 (Zhao et al., 2025b), which introduces a variant of GRPO tailored to dLLMs, and DiffuCoder (Gong et al., 2025), which focuses on enhancing the coding abilities of LLaDA-style models using RL. Most recent extensions aim to improve the quality of policy gradient estimators (Tang et al., 2025; Wang et al., 2025a; Lin et al., 2025; Rojas et al., 2025; Zhu et al., 2025; Wang et al., 2025e; Zhan, 2026), and have demonstrated promising results in further enhancing the reasoning capabilities of dLLMs. The key distinction from our approach is that these methods use a fixed sampling strategy (e.g., high-confidence sampling), and the policy corresponds to the dLLM itself (or a LoRA-augmented version for efficiency). Closest to our work are DCOLT (Huang et al., 2025b), which trains a separate unmasking module (with a fixed number of unmasked tokens in each step) in addition to updating the base model via RL, and DiFFPO (Zhao et al., 2025a), which jointly learns an unmasking confidence threshold while updating the base model through RL. Differently to both these works, our primary goal is not in improving the reasoning abilities of the underlying dLLM; hence we keep the underlying dLLM fixed and focus solely on training a standalone policy with the aim of learning fast and adaptive sampling while preserving the performance of the base model. Another related concurrent work is Hong et al. (2025a) where an unmasking policy for dLLMs trained via GRPO is proposed, similar to our approach. However, their policy unmasks a fixed number of tokens at each step (similar to DCOLT), whereas ours dynamically adapts the number of tokens to unmask via our Bernoulli formulation. Finally, Seed Diffusion (Song et al., 2025) identifies computation-aware reinforcement learning as a key ingredient for achieving competitive efficiency in closed-source coding dLLMs.

**Orthogonal efforts to accelerate dLLMs.** Besides the aforementioned heuristic-based sampling innovations, other efforts to improve inference efficiency in dLLMs (Ma et al., 2025) include KV caching (Jiang et al., 2025; Wu et al., 2025), (variants of) speculative decoding (Israel et al., 2025; Campbell et al., 2025; Guo & Ermon, 2025; Wu & Zhang, 2025), training separate decoder modules during pretraining (Liu et al., 2025a; Arriola et al., 2025b), and diffusion forcing (Wang et al., 2025d), among others. Concurrent work explores distilling faster generation patterns directly into the base model (Chen et al., 2025b), or training a (small) separate unmasking network on top of the pretrained dLLM (Bansal & Sanghavi, 2025; Bao et al., 2025). Crucially, unlike our approach, where this training is done via RL, these works train unmasking modules by distilling generation trajectories from the base model. As a result, we expect that such approaches may still inherit limitations observed in dLLM sampling, such as difficulties in the non semi-AR regime (cf. Section 2.2).

**RL for Adaptive Compute** Since our sampling policies result in a variable number of sampling steps $(T - \hat{T})$ per input, our work connects to the broader literature on adaptive computation (Graves, 2016; Teerapittayanon et al., 2016). Notable examples of using RL to learn input-dependent compute policies include conditional computation with stochastic gating (Bengio et al., 2015), dynamic block skipping in residual networks (Wang et al., 2018), and RL-based early-exit decisions for long chain-of-thought reasoning (Dai et al., 2025). To the best of our knowledge, our work is the first to explore learning adaptive policies using RL for the task of sampling in dLLMs. The closest related concurrent work is DiFFPO (Zhao et al., 2025a); however, unlike our approach–where the policy is learned end-to-end–DiFFPO learns to predict adaptive thresholds, which are then used in the same fashion as the fixed thresholds employed by Fast-dLLM (Wu et al., 2025). Concurrently, there is also a growing body of work that uses RL to introduce adaptivity into reasoning chains of autoregressive LLMs (Arora & Zanette, 2025; Yi et al., 2025).

# B. Policy Sampling Algorithm

*Algorithm 1.* Policy Sampling (full-diffusion setting $BL = L$)

1: **Input:** Prompt $\boldsymbol{x} \in \mathcal{V}^d$, maximum generation length $L$, maximum diffusion steps $T$, dLLM $p_\theta$, sampling policy $\pi_\phi$ (via $f_\phi$), policy temperature $\tau_\pi$
2: **Output:** Generated sequence $\boldsymbol{y}_{\hat{T}}$, number of sampling steps
3: $\boldsymbol{y}_T \leftarrow (\boldsymbol{M}, \ldots, \boldsymbol{M})$ $\qquad\qquad\qquad$ ▷ Fully masked initial sequence
4: $\mathcal{M}_T \leftarrow \{1, \ldots, L\}$ $\qquad\qquad\qquad$ ▷ Set of still-masked positions
5: **for** $t = T, T-1, \ldots, 1$ **do**
6: $\quad$ **if** $\mathcal{M}_t = \emptyset$ **then**
7: $\quad\quad$ **break** $\qquad\qquad\qquad$ ▷ Stop generating when all unmasked
8: $\quad$ **end if**
9: $\quad p_t^k \leftarrow p_\theta(\cdot \mid \boldsymbol{x}, \boldsymbol{y}_t), \forall k \in [L]$ $\qquad\qquad$ ▷ Token distributions
10: $\quad c_t^k \leftarrow \max_{v \in \mathcal{V}} p_t^k(v), \forall k \in [L]$ $\qquad\qquad$ ▷ Token confidences
11: $\quad \boldsymbol{c}_t \leftarrow (c_t^1, \ldots, c_t^L)$
12: $\quad m_t^k \leftarrow \mathbf{1}[k \in \mathcal{M}_t], \forall k \in [L]$
13: $\quad \boldsymbol{b}_t \leftarrow f_\phi(\boldsymbol{c}_t, \boldsymbol{m}_t, t)$ $\qquad\qquad\qquad$ ▷ Policy logits
14: $\quad s_t^k \leftarrow \sigma(b_t^k / \tau_\pi), \forall k \in \mathcal{M}_t$
15: $\quad u_t^k \sim \text{Ber}(s_t^k), \forall k \in \mathcal{M}_t$ $\qquad\qquad$ ▷ Unmasking decisions
16: $\quad u_t^k \leftarrow 0, \forall k \notin \mathcal{M}_t$ $\qquad\qquad$ ▷ Ignore already-unmasked
17: $\quad$ **if** $\sum_{k=1}^{L} u_t^k = 0$ **then**
18: $\quad\quad u_t^{k^*} \leftarrow 1, k^* = \text{argmax}_{k \in \mathcal{M}_t} s_t^k$ $\qquad$ ▷ Argmax fallback
19: $\quad$ **end if**
20: $\quad \mathcal{U}_t^\pi \leftarrow \{k \in \mathcal{M}_t \mid u_t^k = 1\}$
21: $\quad y_{t-1}^k \sim p_t^k, \forall k \in \mathcal{U}_t^\pi$ $\qquad\qquad\qquad$ ▷ Sample tokens
22: $\quad y_{t-1}^k \leftarrow y_t^k, \forall k \notin \mathcal{U}_t^\pi$
23: $\quad \mathcal{M}_{t-1} \leftarrow \mathcal{M}_t \setminus \mathcal{U}_t^\pi$
24: **end for**
25: $\hat{T} \leftarrow t$
26: **return** $\boldsymbol{y}_{\hat{T}}, T - \hat{T}$

# C. Additional Results

## C.1. Figure 1 replicated for {LLADA-8B-INSTRUCT, DREAM-7B-INSTRUCT} × {GSM8K, MATH-500}

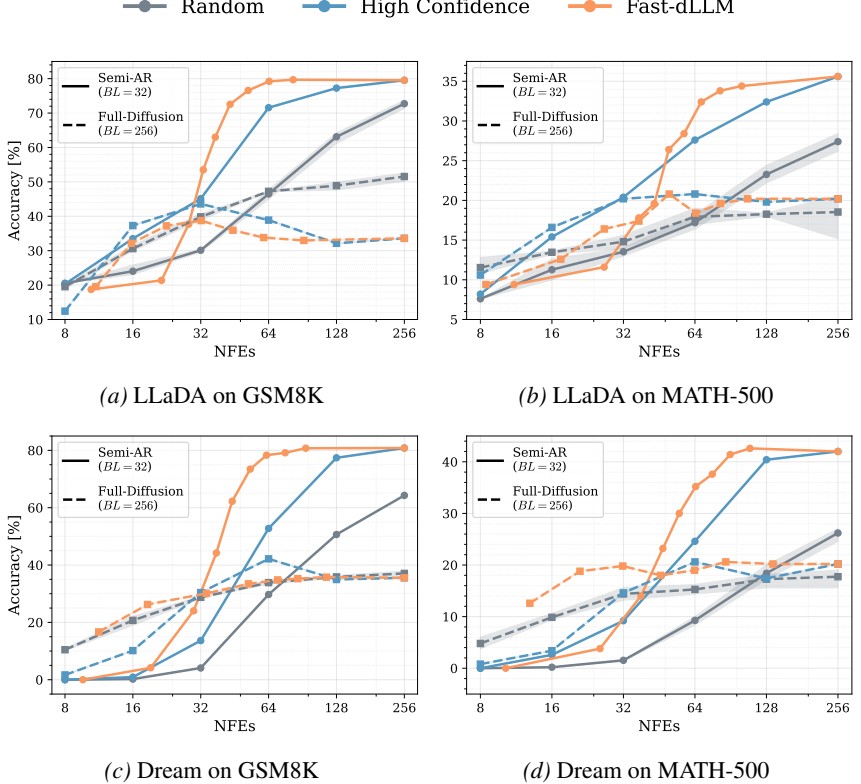

*(a)* LLaDA on GSM8K

*(b)* LLaDA on MATH-500

*(c)* Dream on GSM8K

*(d)* Dream on MATH-500

*Figure 10.* Performance comparison with ($BL = 32$; —) and without ($BL = 256$; - -) semi-AR generation. The same trend observed in Figure 1 holds across all models and datasets: confidence-based heuristics perform well under semi-AR generation but degrade significantly without it.

## C.2. Figure 4 replicated using additional baselines

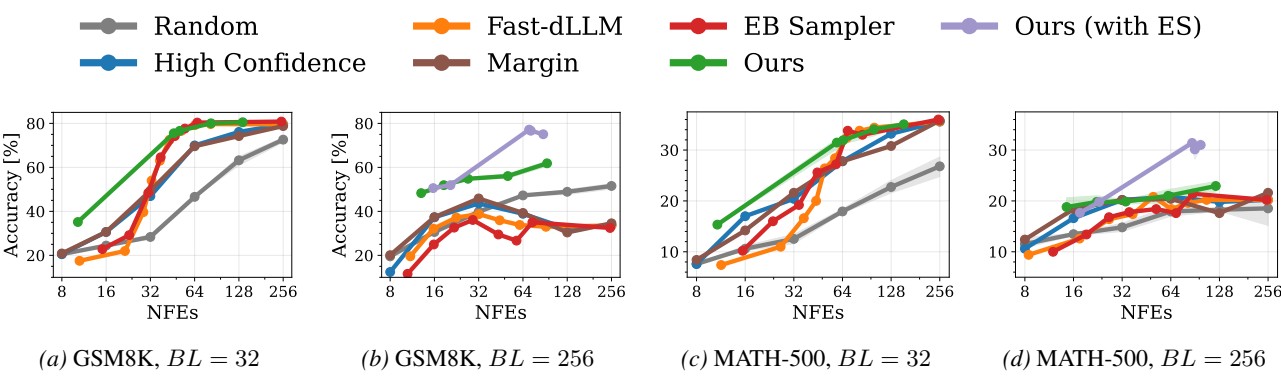

*(a)* GSM8K, $BL = 32$

*(b)* GSM8K, $BL = 256$

*(c)* MATH-500, $BL = 32$

*(d)* MATH-500, $BL = 256$

*Figure 11.* Figure 4 reproduced using additional baselines from (Kim et al., 2025a) and (Ben-Hamu et al., 2025). We observe our main findings still hold: our policies mainly match the performance of the best-performing baselines under the semi-AR regime ($BL = 32$) while outperforming them in the full-diffusion setting ($BL = 256$).

## C.3. Figure 4 replicated using wall-clock time as the efficiency measure

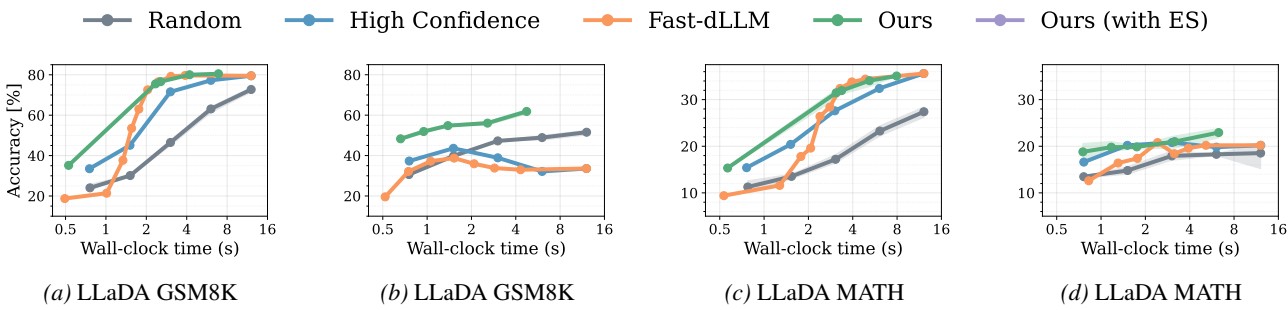

*(a) LLaDA GSM8K*  *(b) LLaDA GSM8K*  *(c) LLaDA MATH*  *(d) LLaDA MATH*

*Figure 12.* Figure 4 reproduced using wall-clock time (in seconds) as the efficiency metric instead of NFEs. The difference in our policy curves (—) when measured in wall-clock time versus NFEs (see Figure 4) is minimal to non-existent, demonstrating the negligible computational overhead of our policy. This low overhead is primarily due to the small size of the unmasking model relative to the base dLLM (300K vs. 8B parameters in the case of LLaDA). All experiments run on A100 GPUs.

## C.4. Figure 4 replicated for DREAM-7B-INSTRUCT

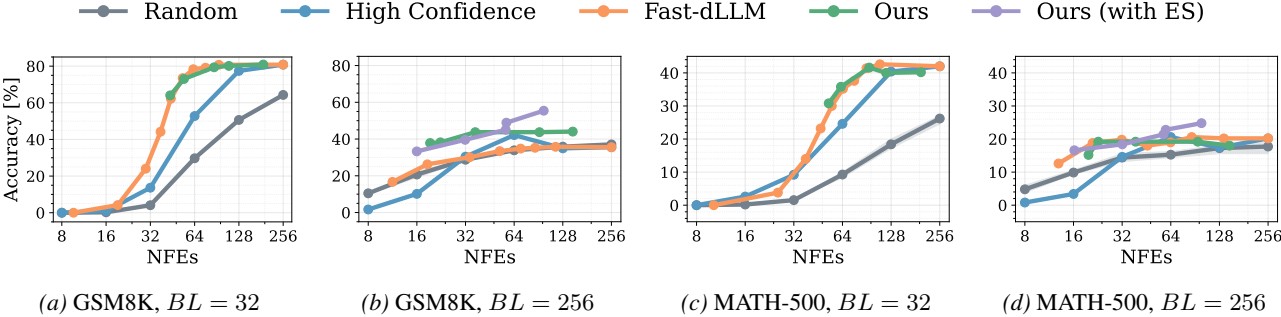

*(a) GSM8K, $BL = 32$*  *(b) GSM8K, $BL = 256$*  *(c) MATH-500, $BL = 32$*  *(d) MATH-500, $BL = 256$*

*Figure 13.* Results for Dream in semi-AR (Figure 13a & Figure 13c) and full-diffusion (Figure 13b & Figure 13d) generation regimes. For the policies we vary $\alpha \in \{10, 3, 1, 0.3, 0\}$ and use $\tau_\pi = 0.5$ for $BL = 32$ and $\tau_\pi = 1$ for $BL = 256$.

## C.5. Controllability of learning unmasking policies with RL (via $\alpha$)

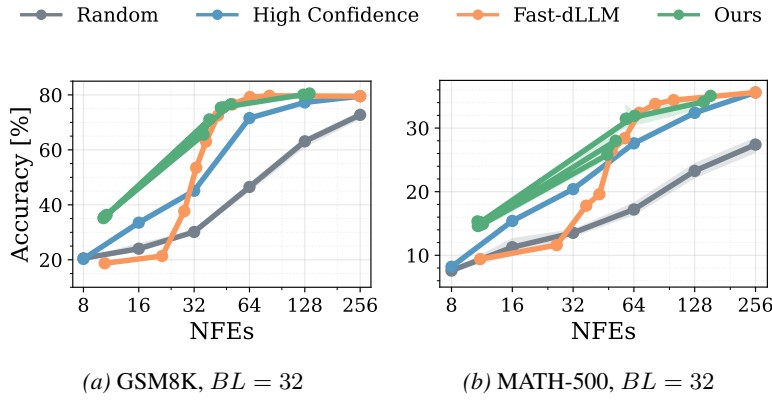

*(a) GSM8K, $BL = 32$*  *(b) MATH-500, $BL = 32$*

*Figure 14.* $BL = 32$ results for LLaDA with a denser regularization grid, $\alpha \in \{10.0, 9.0, \ldots, 1.0, 0.3, 0.0\}$. Single training seed due to cost; error bars show (min, max) over three test-time seeds. Note that for $\alpha \geq 4.0$, the change in NFEs is not monotonic; different values lead to convergence either close to the $\alpha = 3.0$ policy, or to that of $\alpha = 10.0$. Points for policy results are connected by $\alpha$ to emphasize the non-monotone behavior.

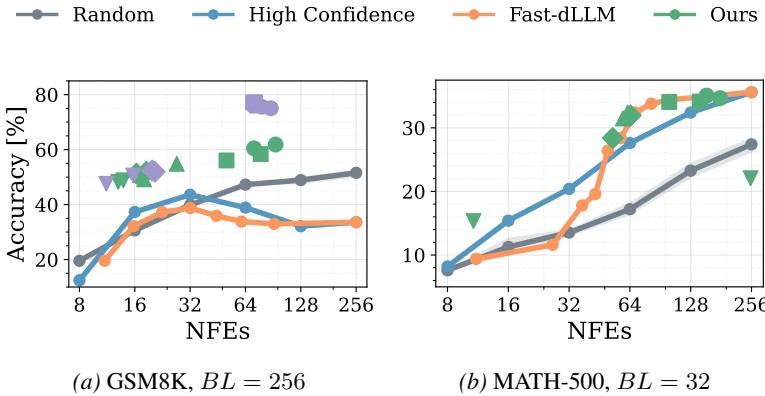

*(a)* GSM8K, $BL = 256$      *(b)* MATH-500, $BL = 32$

*Figure 15.* $BL = 256$ (GSM8K) and $BL = 32$ (MATH-500) results for LLaDA with all training seeds scattered. Even for a fixed value of $\alpha$, the resulting policy can vary in accuracy and speed due to the randomness of the training procedure. Marker shape denotes the value of $\alpha$.

## C.6. Impact of policy temperature $\tau_\pi$

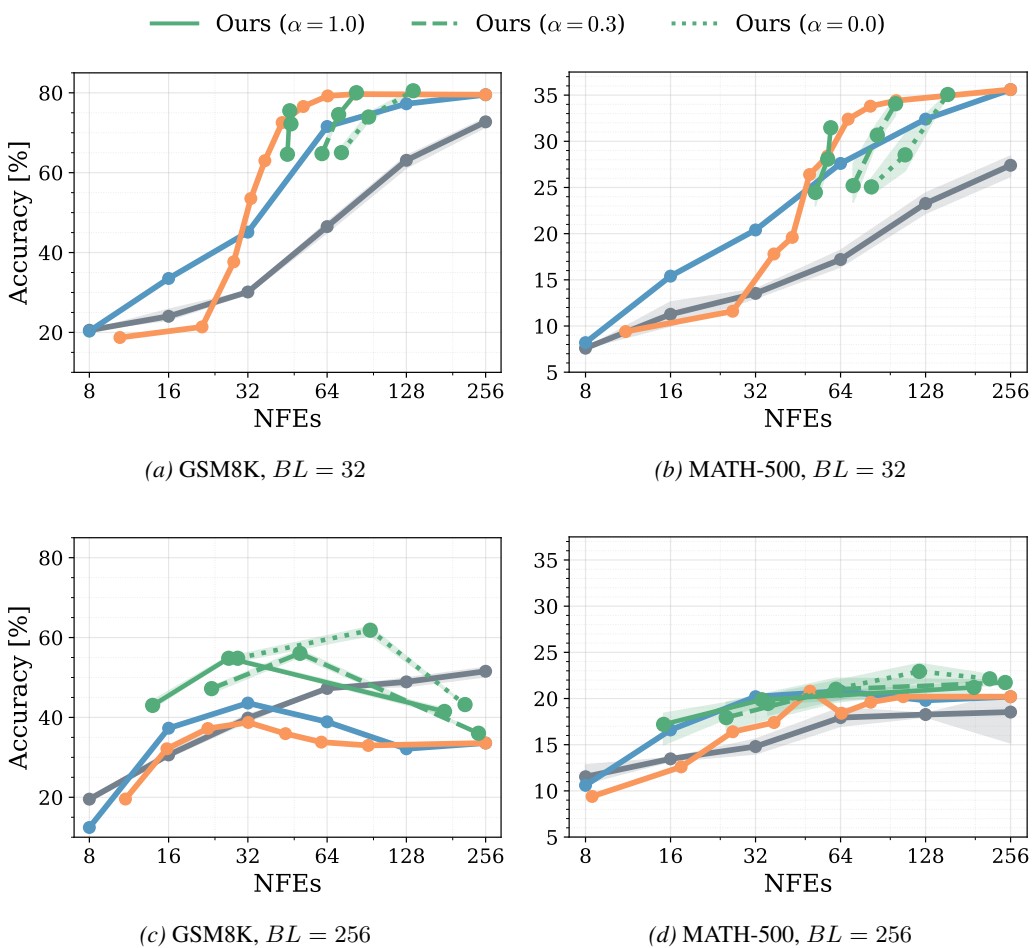

*(a)* GSM8K, $BL = 32$          *(b)* MATH-500, $BL = 32$

*(c)* GSM8K, $BL = 256$          *(d)* MATH-500, $BL = 256$

*Figure 16.* We study the effect of changing the policy temperature $\tau_\pi$ (cf. Section 3.2). For each $\alpha \in \{3, 0.3, 0\}$, we construct a corresponding test-time Pareto frontier by varying $\tau_\pi \in \{1.5, 1.0, 0.5\}$. Interestingly, in some cases—such as $\alpha = 0$ with $BL = 32$—adjusting $\tau_\pi$ enables an effective trade-off between compute and performance. Moreover, we find that $\tau_\pi = 0.5$ is optimal in the semi-AR, while $\tau_\pi = 1$ performs best in the full-diffusion ($BL = L = 256$) setting.

## C.7. Model transfer results

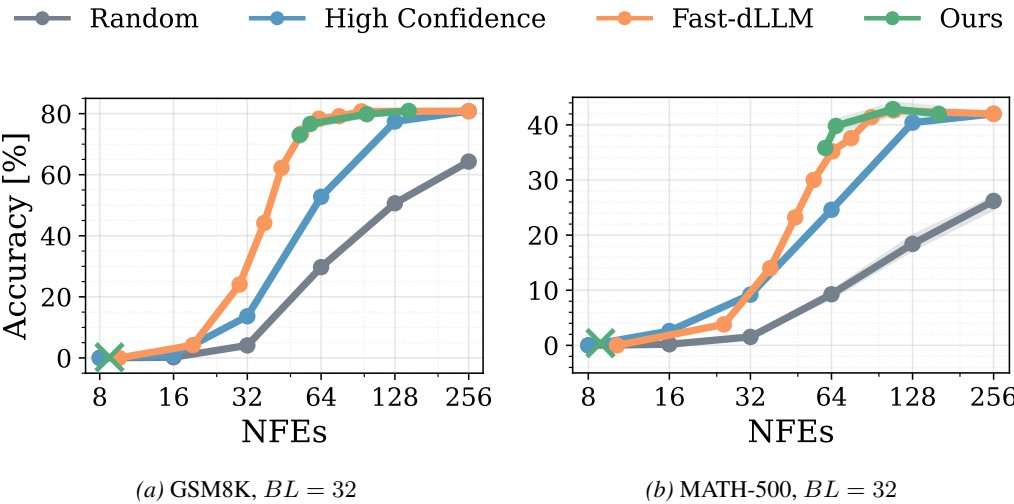

*(a) GSM8K, $BL = 32$*         *(b) MATH-500, $BL = 32$*

*Figure 17.* Model transfer results. We use policies trained on LLaDA and evaluate them on Dream with $\tau_\pi = 0.5$. Encouragingly, transferred policies give comparable results to training Dream specific policies (cf. Figure 13a & Figure 13c). Note that the $\alpha = 10$ policy is represented separately by a (✖) in the lower left to avoid misleading visualization when interpolating to $\alpha = 3$.

## C.8. Sequence-length transferability

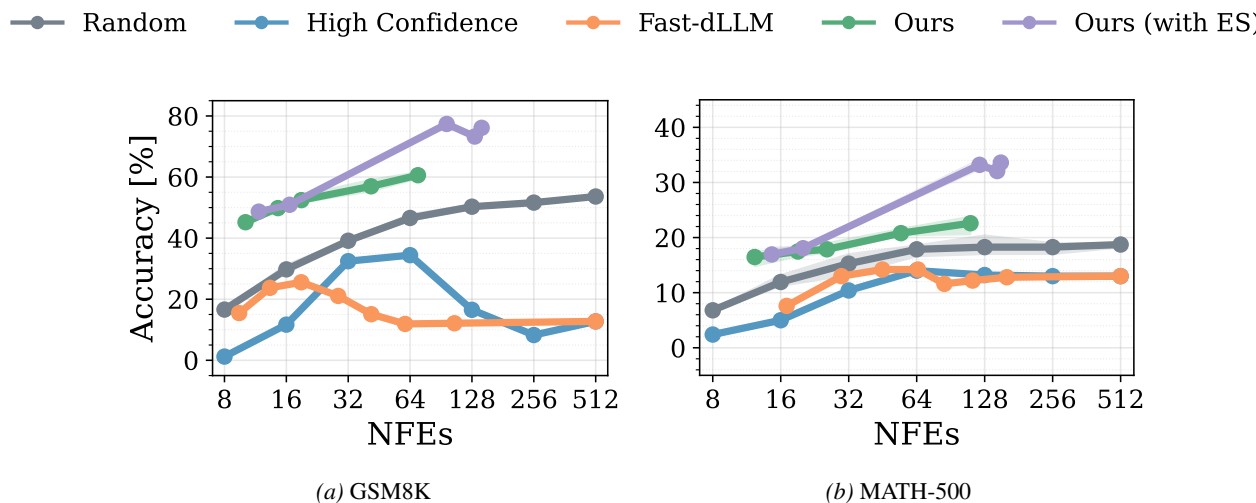

*(a) GSM8K*         *(b) MATH-500*

*Figure 18.* $BL = L = 256$-trained policies from Section 4.2 evaluated with a 2x longer sequence length ($BL = L = 512$) with $\tau_\pi = 1$. Note that the learned policies yield almost identical performance, while the heuristic methods degrade further compared to $L = 256$ (cf. Figure 4b & Figure 4d). Both for LLaDA-8B-Instruct.

## C.9. Non-greedy decoding results

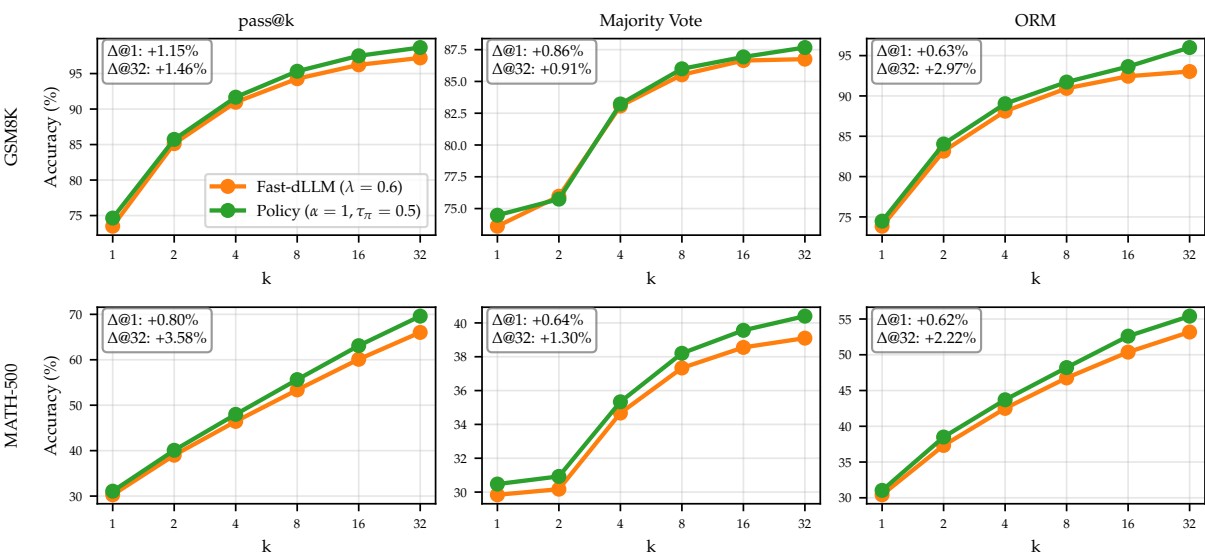

*Figure 19.* We compare our policy ($\alpha = 1$) sampling with Fast-dLLM ($\lambda = 0.6$) in $BL = 32$ setting under non-greedy decoding (using temperature $\tau = 0.8$). *Left*: we observe that policy sampling exhibits better scaling in terms of pass@k, indicating that additional stochasticity due to Bernoulli sampling of unmasking decisions can help with exploring the solution space compared to "deterministic" Fast-dLLM. *Middle*: when the final answer is picked based on majority vote (i.e., self-consistency) policy sampling outperforms Fast-dLLM. *Right*: when then final answer is picked using an outcome reward model (ORM; Liu et al. 2025b) instead, the policy sampling maintains its edge over Fast-dLLM. For GSM8K, we use a subset of $N_{\text{test}} = 300$ samples to ease computational burden.

## C.10. Additional policy input ablations

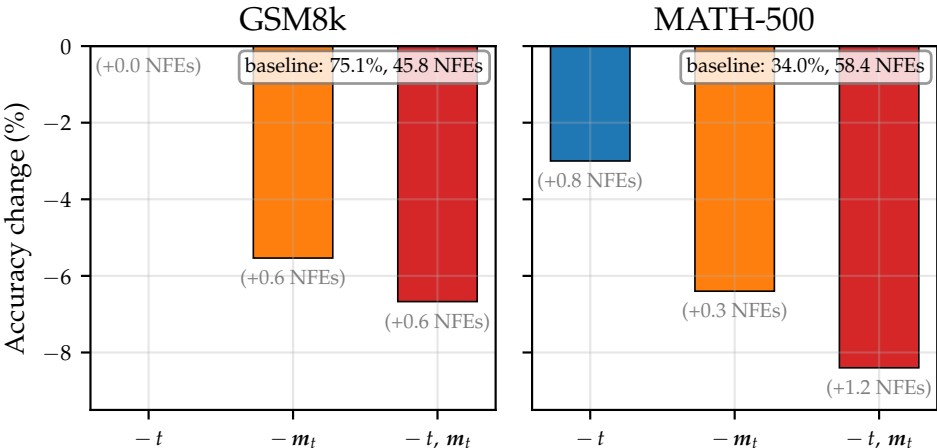

*Figure 20.* We perform (test-time) ablations on the policy inputs during generation. We use policy trained with $\alpha = 1$ on LLaDA with $BL = 32$. Concretely, we set to zero either i) time ($t$), ii) masks ($m_t$), or iii) both. We report drops in accuracy as (negative) bars and display the change in average NFEs as text underneath each bar. We observe that zeroing out part of the input leads to reduced performance in all cases, except when zeroing out only time on GSM8K. Interestingly, zeroing out time or masks seems to have a minimal impact on the speed of the policy (speed changes at most for 1.2 NFEs on average).

# D. What are RL unmasking policies actually doing?

To better understand the behaviour of our learned policies, we conduct a qualitative analysis comparing generation/unmasking trajectories produced by our RL policies against a confidence heuristic like Fast-dLLM (Wu et al., 2025). We consider three settings that span different points on the accuracy-efficiency Pareto frontier and different generation regimes (see Figure 21):

- **BL32-slow** (⬭): Semi-AR ($BL = 32$) with Fast-dLLM ($\lambda = 0.9$) vs. Policy ($\alpha = 0.3$)

- **BL32-fast** (◯): Semi-AR ($BL = 32$) with Fast-dLLM ($\lambda = 0.1$) vs. Policy ($\alpha = 10$)

- **BL256** (⬭): Full-diffusion ($BL = L = 256$) with Fast-dLLM ($\lambda = 0.9$) vs. Policy ($\alpha = 0.3$) vs. Policy with expert steering ($\alpha = 0.3$)

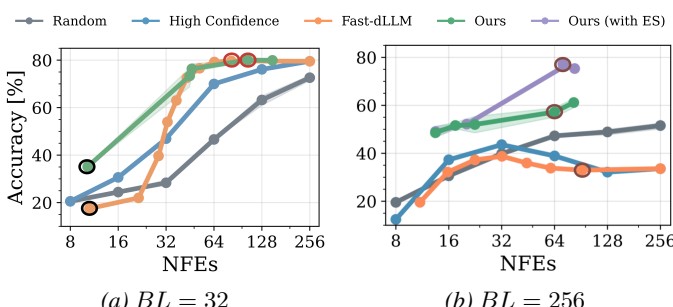

*(a) $BL = 32$*       *(b) $BL = 256$*

*Figure 21.* Results for LLaDA in semi-AR ($BL = 32$; replicated from Figure 4a) and full-diffusion ($BL = 256$; figure replicated from Figure 4b) generation regimes. We highlight the policies used in our qualitative analysis of unmasking orders: ⬭ for **BL32-slow**, ◯ for **BL32-fast**, and ⬭ for **BL256**.

Given a prompt $x$ and generated answer $\hat{y}$ (with a maximum answer length and maximum number of steps to set to $L$ and $T$, respectively), the *unmasking order*[3] is defined as $o \in [T]^L$, where $o_k := \min\{t \mid y_t^k \neq M\}$ denotes the timestep when the token at position $k$ is first unmasked. For presentation clarity, we reverse the time axis relative to the main paper, so time flows as $t = 1, 2, \ldots, T$. For all experiments, we use $N = 100$ GSM8k test samples with $L = T = 256$.

We describe main insights for each setting next.

## D.1. BL32-slow

In this setting, our policy achieves performance nearly identical to Fast-dLLM in both accuracy and efficiency (see ⬭ in Figure 21a). Despite these similarities, we find that the learned policy adopts a somewhat different sampling strategy compared to plain confidence thresholding. This is visually apparent in Figure 23, where we observe that, although both strategies predominantly follow a left-to-right generation pattern (due to semi-AR regime), the policy's unmasking order lines (—) display more "wiggliness" within each block. This is due to the fact that, as shown in Figure 24, the policy is far less likely to unmask adjacent tokens simultaneously—doing so for only around $\sim 20\%$ of tokens, in contrast to Fast-dLLM, which does so for $\sim 65\%$ on average. This suggests that the policy learns a more dispersed unmasking strategy, favoring token commit-

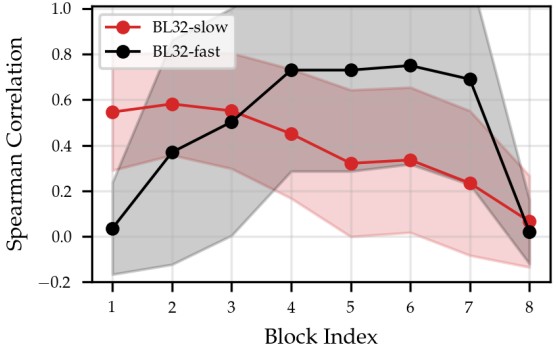

*Figure 22.* Spearman rank correlation between the unmasking orders of Fast-dLLM and our RL policies per block for semi-AR policies considered here. Note that for certain blocks, the unmasking orders exhibit zero correlation: first and last block for BL32-fast(◯), and last block for BL32-slow (⬭).

ments at scattered positions within each block rather than in contiguous chunks. Such a scattered approach may enable the model to gather more diverse contextual information before finalizing neighboring tokens, potentially mitigating errors that arise from parallel generation. The more scattered generation pattern within each block is also evident in the token generation trajectories shown in Figure 27.

The two methods also differ substantially in how they allocate compute across blocks. As shown in Figure 5a, Fast-dLLM assigns a larger number of NFEs to earlier blocks, allocating $\sim 17$ NFEs to the first block and gradually decreasing to just $\sim 4$ NFEs in the final block. In contrast, our learned policy distributes compute almost uniformly across blocks, using $\sim 10$ NFEs per block. This likely also contributes to the decreasing Spearman rank correlation of unmasking orders between the policy and Fast-dLLM sampling across blocks (see Figure 22).

---

[3]Note that we consider partial orders here, as multiple tokens can be unmasked simultaneously.

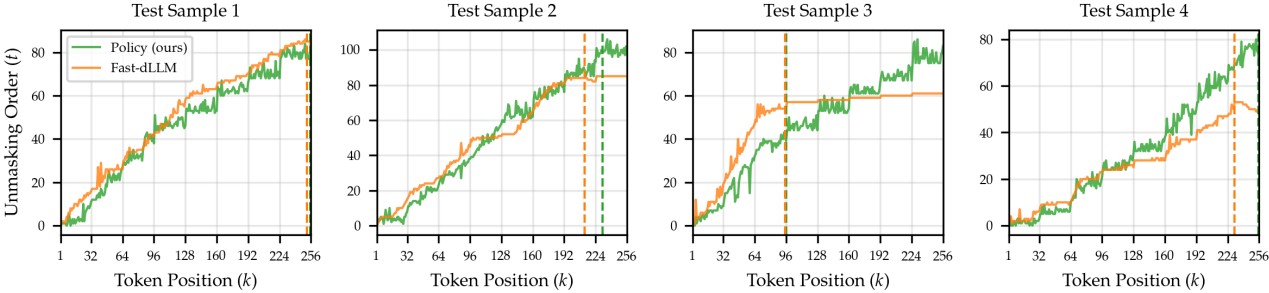

*Figure 23.* **BL32-slow**: unmasking orders for policy sampling ($o_{\alpha=0.3}$) and Fast-dLLM ($o_{\lambda=0.9}$) for four GSM8k test samples under semi-AR generation ($BL = 32, L = 256$). Vertical dashed lines indicate the position where the model outputs the end-of-sequence (*EOS*) token. Interestingly, Fast-dLLM appears more effective at accelerating generation after producing the *EOS* token, whereas policy sampling tends to sometimes waste compute by not unmasking all padding tokens (within the current block) after EOS in parallel, see the third test sample.

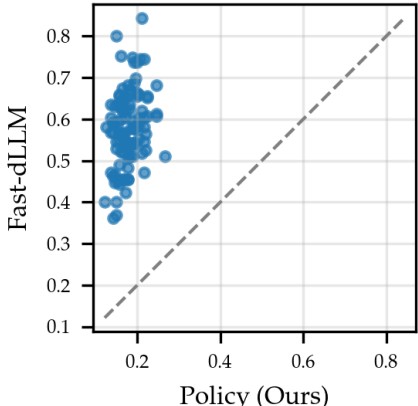

*Figure 24.* **BL32-slow**: for each of the $N = 100$ samples, we report the frequency of tokens whose right-adjacent token was unmasked at the same sampling step under Fast-dLLM versus our policy sampling. Interestingly, policy sampling unmasks adjacent tokens much less frequently than Fast-dLLM.

## D.2. BL32-fast

Here both strategies achieve very fast but low-accuracy sampling (see ◯ in Figure 21a). However, policy sampling significantly outperforms Fast-dLLM, achieving approximately $\sim 38\%$ accuracy compared to $\sim 18\%$. Examining the unmasking orders in Figure 25 and the average NFEs per block in Figure 5b, we observe that policy sampling tends to slow down in the final block, typically generating the numerical answer last (see also the token generation trajectory in Figure 29 and how the tokens of the numerical answer are generated last under policy sampling). This behavior may help explain its improved performance over Fast-dLLM and shows the promise of RL for discovering new sampling techniques.

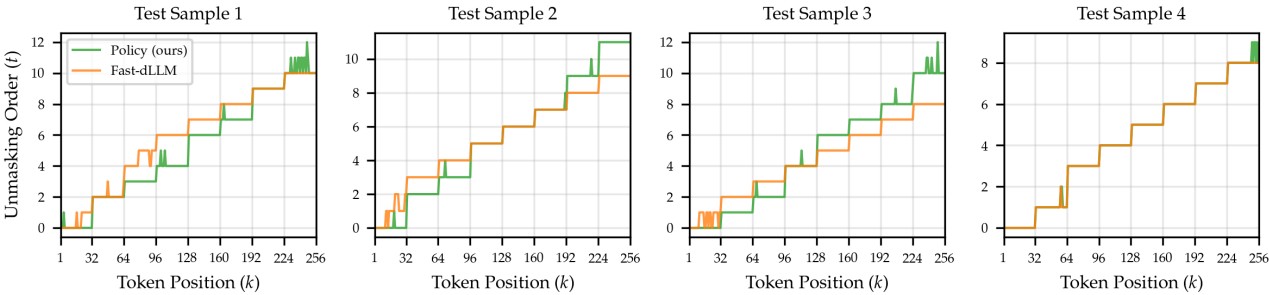

*Figure 25.* **BL32-fast**: unmasking orders for policy sampling ($o_{\alpha=10}$) and Fast-dLLM ($o_{\lambda=0.1}$) for four GSM8k test samples under semi-AR generation ($BL = 32, L = 256$). Under low-NFE constraints, both strategies often predict all tokens within a block simultaneously (as indicated by flat unmasking-order lines). In the final block, when generating the numerical answer, the policy slows down.

## D.3. BL256

Lastly, we consider the full-diffusion setting ($BL = L = 256$). While all three strategies achieve a comparable number of NFEs (see ◯ in Figure 21b), their accuracy differs significantly. The fact that these strategies implement notably different sampling procedures is further evidenced by the low values of Spearman rank correlations $0.26 \pm 0.21$ when comparing Fast-dLLM orders with policy sampling, and $-0.24 \pm 0.51$ when comparing Fast-dLLM with policy trained with expert steering. These differences are also reflected in the divergent unmasking orders shown in Figure 26 (also refer to Figure 7 for a plot of average unmasking order across samples).

Starting with Fast-dLLM (—), we observe in Figure 26 that it often generates tokens in a "reverse" order (right-to-left), see also the token trajectories in Figure 32). This behavior likely stems from the fact that LLaDA did not exclude padding tokens from the loss computation during SFT (Nie et al., 2025) to preserve the dLLM's ability to generate sequences of varying (effective) lengths. Consequently, the model overfits to padding tokens, which confidence-based methods like Fast-dLLM then mistakenly prioritize, leading to their early generation. This helps explain the poor performance of confidence-based strategies observed in Figure 1, highlighting the fragility of heuristic-driven approaches.

Looking at the unmasking orders of policy sampling (—), the policy appears to implement a near-random unmasking strategy, as there is little evident spatial structure in the unmasking order. This is encouraging, as it suggests that the policy can learn to disregard corrupted signals—such as inflated confidence for padding tokens—without explicit supervision, leading to more robust performance across different generation settings (semi-AR vs full-diffusion). Lastly, we observe that the policy with expert steering (—) generates in an AR fashion (left-to-right). This behavior likely accounts for its superior performance in the full-diffusion setting. It indicates that, during full-diffusion training, the policy is able to pick up the accuracy benefits of AR generation from its expert demonstrations—drawn from the semi-AR regime (Appendix G)—while still learning to generate significantly faster than a purely AR strategy that unmasks one token at a time.

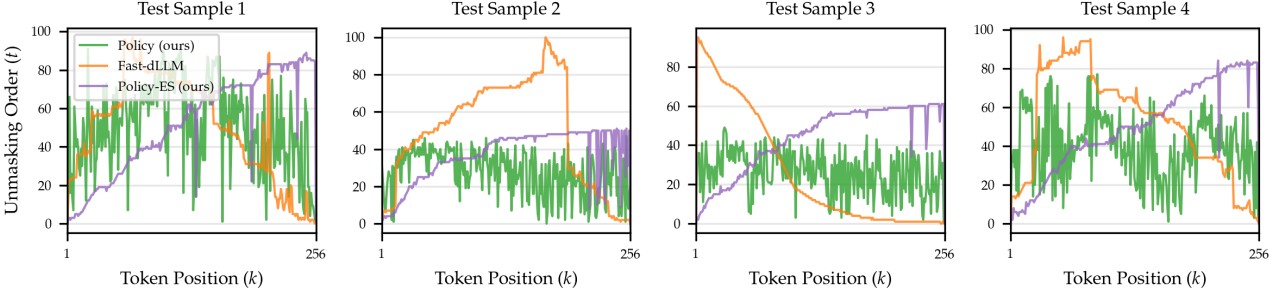

*Figure 26.* **BL256:** unmasking orders for policy sampling ($\boldsymbol{o}_\alpha = 0.3$) and Fast-dLLM ($\boldsymbol{o}_{\lambda=0.9}$) for four GSM8k test samples under full-diffusion generation ($BL = L = 256$).

# E. Generation Trajectories Visualizations

Here we visualize token generation trajectories for the various sampling strategies considered in our work. Specifically, for a randomly selected GSM8k test sample, we display the generated token at each of the $L = 256$ positions, along with its unmasking time (shown in the bottom right corner of each cell, starting from $t = 0$). For visual clarity, token cells are color-coded according to their unmasking time, with blue indicating earlier and red indicating later unmasking. Note that for semi-AR generations, color-coding is done within each block separately.

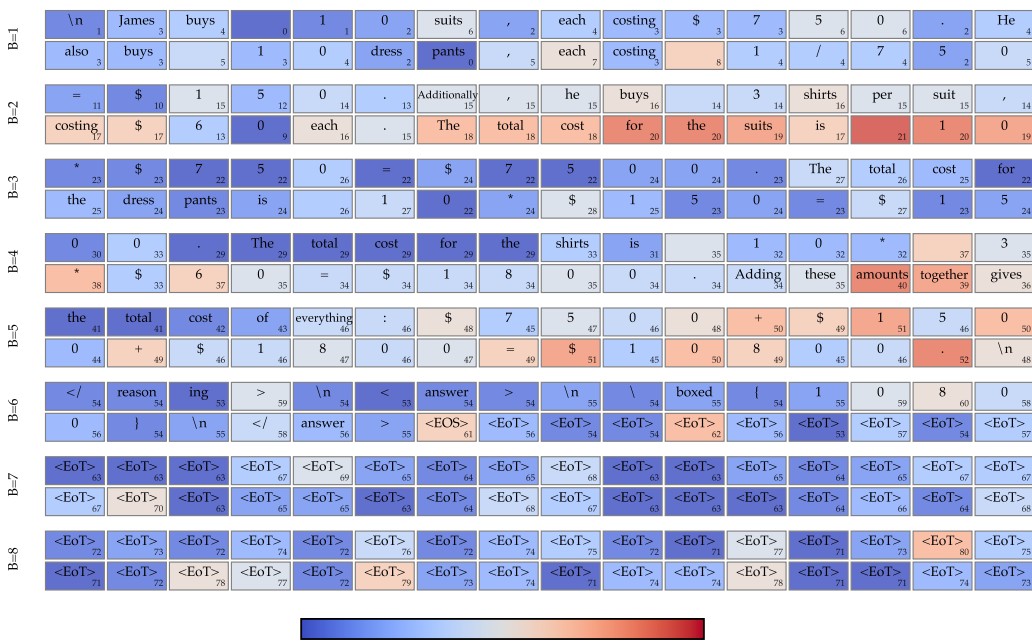

Unmasking Time within Block (early → late)

*Figure 27.* **BL32-slow**: generation trajectory for policy sampling ($\alpha = 0.3$, semi-AR).

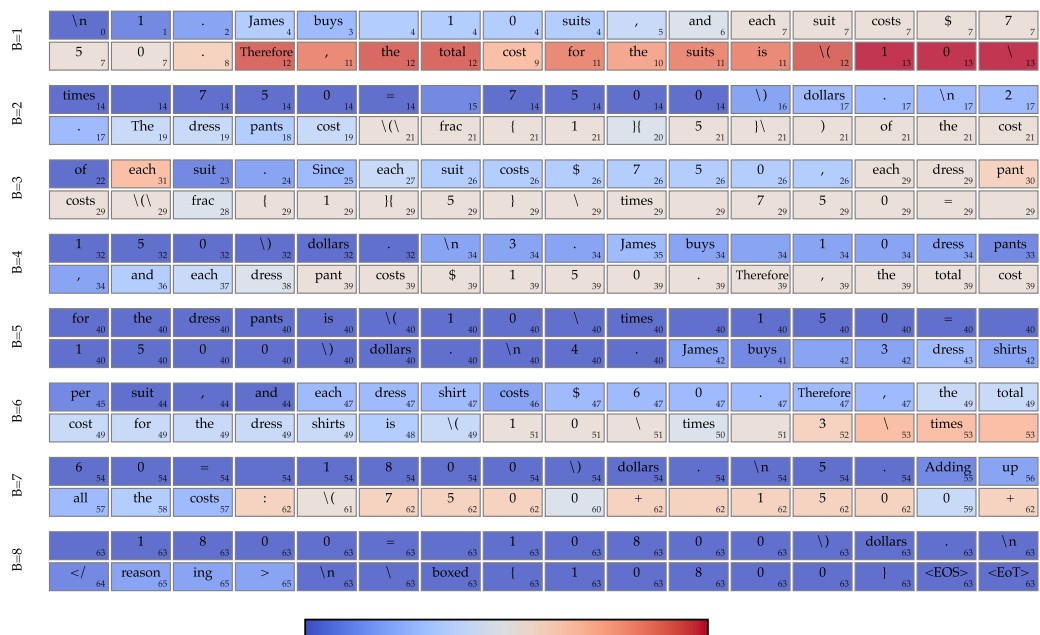

Unmasking Time within Block (early → late)

*Figure 28.* **BL32-slow**: generation trajectory for Fast-dLLM sampling ($\lambda = 0.9$, semi-AR).

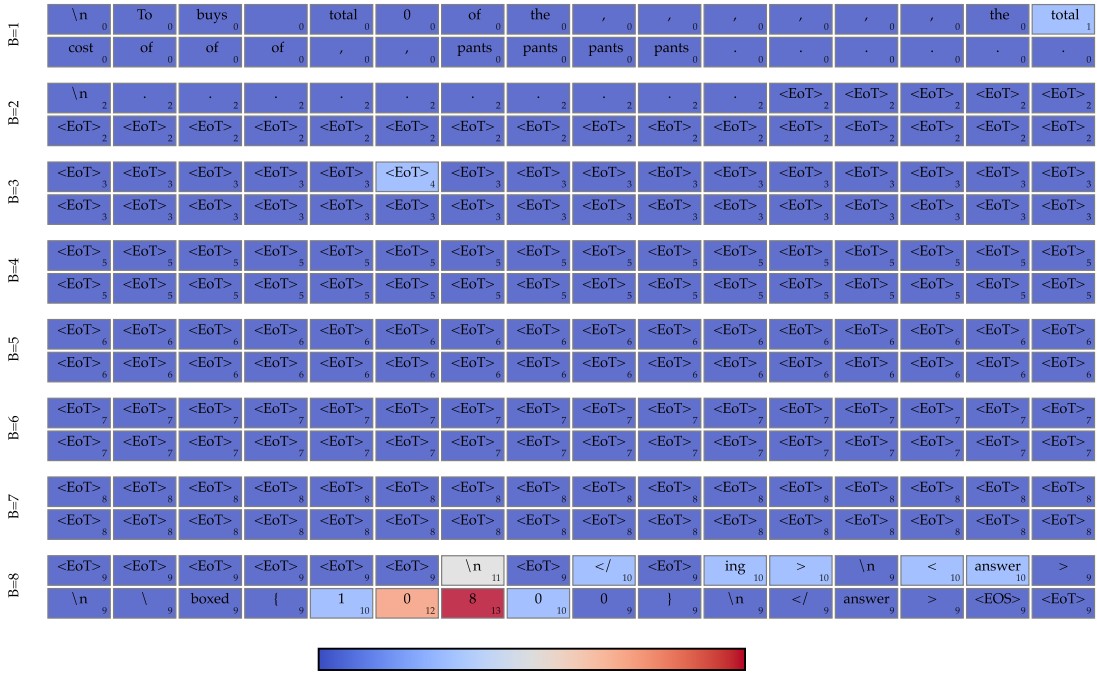

*Figure 29.* **BL32-fast**: generation trajectory for policy sampling ($\alpha = 10$, semi-AR).

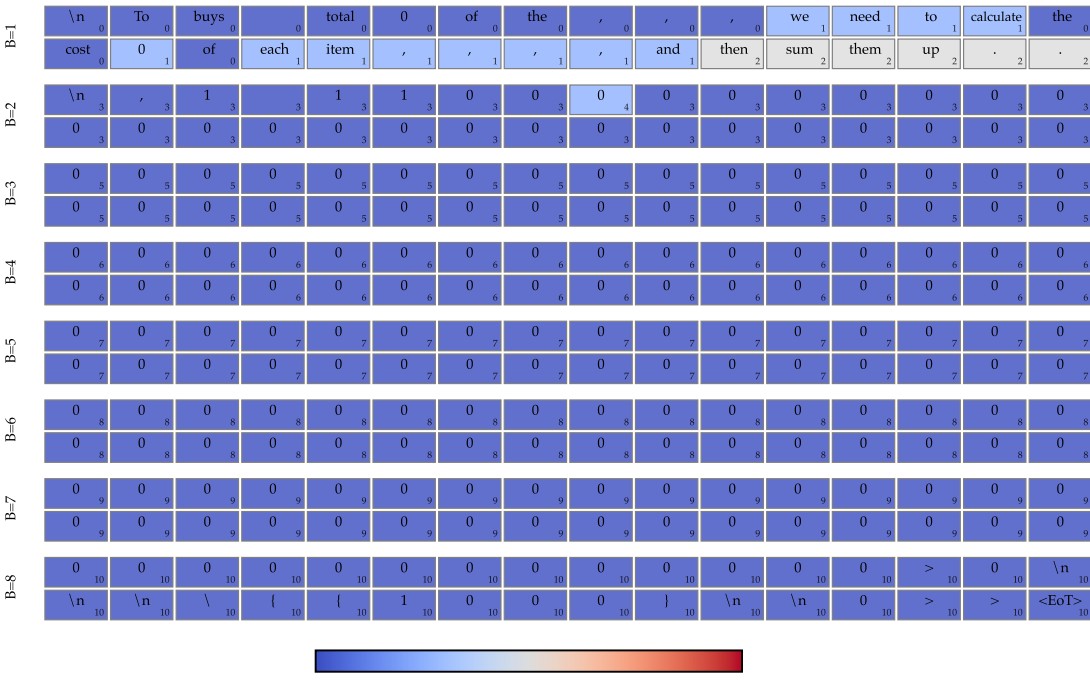

*Figure 30.* **BL32-fast**: generation trajectory for Fast-dLLM sampling ($\lambda = 0.1$, semi-AR).

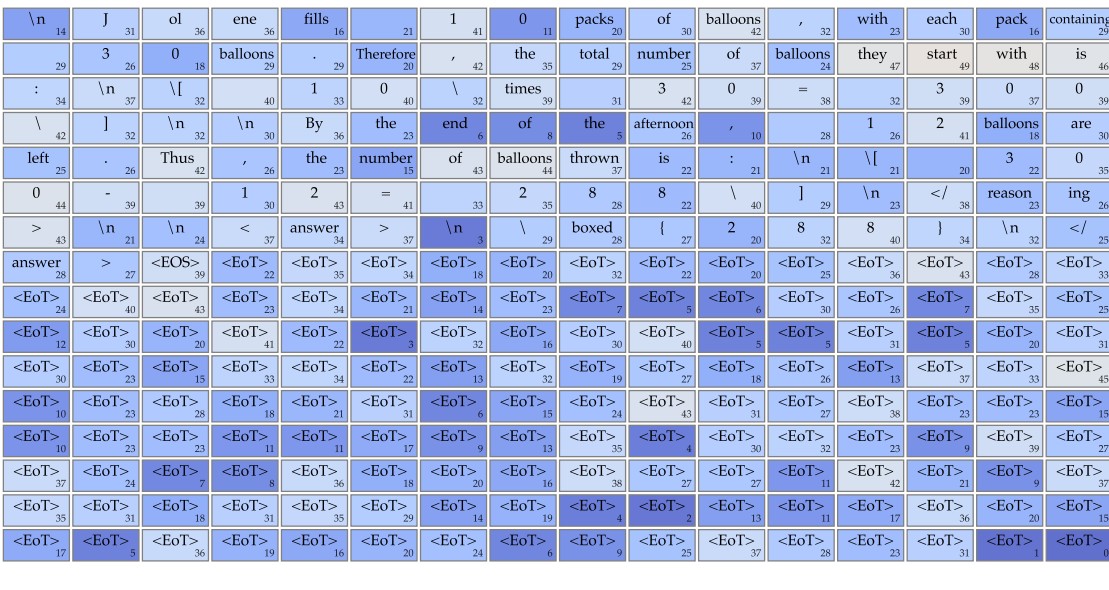

Unmasking Time (early → late)

*Figure 31.* **BL256**: generation trajectory for policy sampling ($\alpha = 0.3$, full-diffusion).

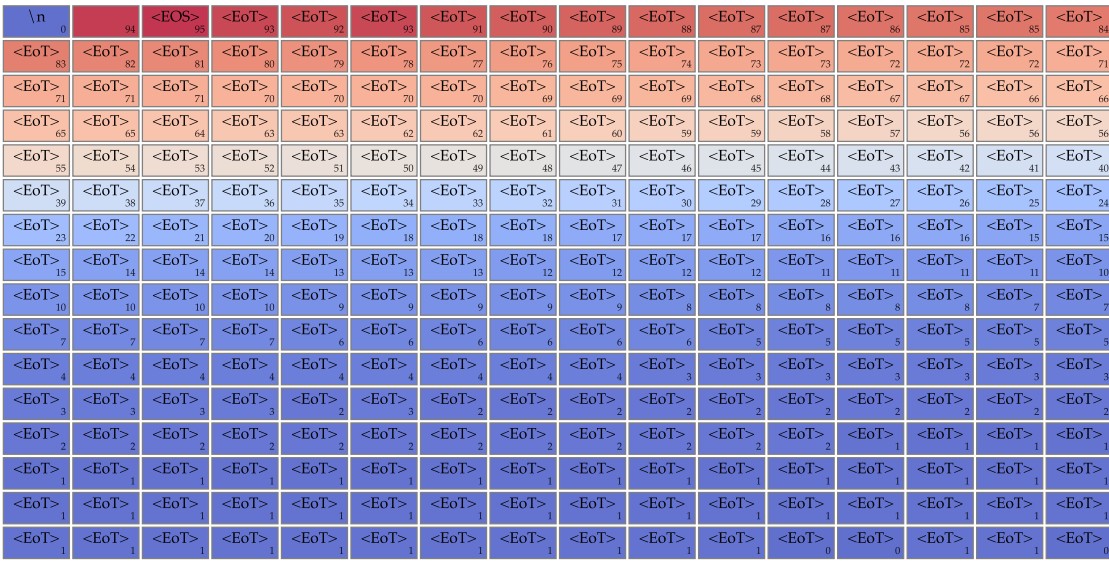

Unmasking Time (early → late)

*Figure 32.* **BL256**: generation trajectory for Fast-dLLM sampling ($\lambda = 0.9$, full-diffusion).

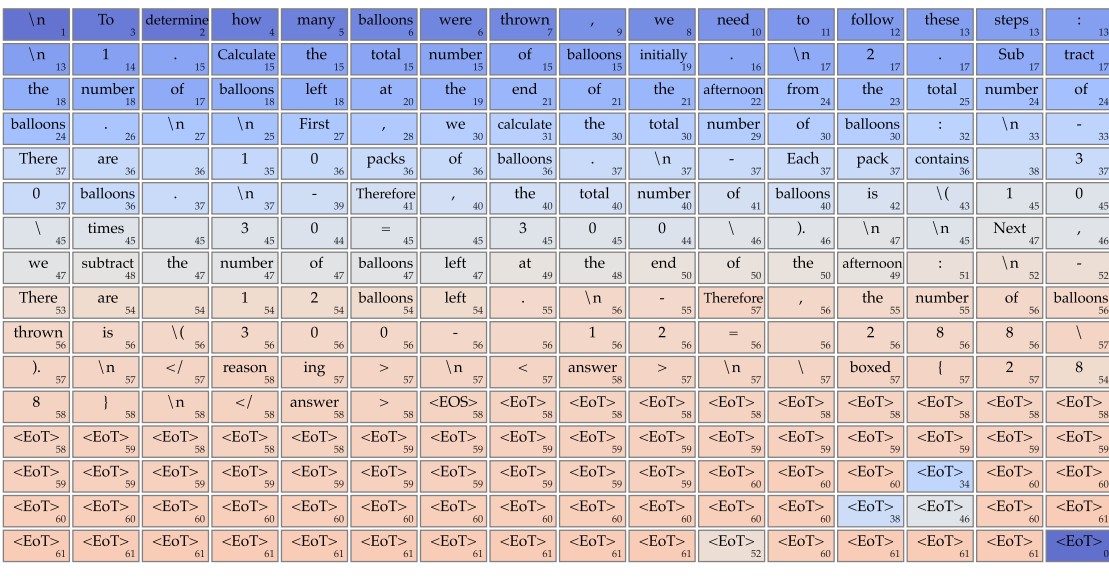

Unmasking Time (early → late)

*Figure 33.* **BL256**: generation trajectory for policy sampling ($\alpha = 0.3$, expert steering, full-diffusion).

# F. Dynamic Plackett-Luce Sampling

We detail here the dynamic Plackett-Luce sampling (DPLS) strategy as an alternative to the Bernoulli sampling (cf. Section 3.2) used in the main experiments of this paper. The name reflects the connection the Plackett-Luce (PL) model proposed by (Luce, 1959) and (Plackett, 1975), with one key difference: the number of selected items is not fixed but can vary freely between $1$ and $L$.

Formally, the Plackett-Luce model operates as follows. Let $\boldsymbol{b}_t = (b_t^1, \ldots, b_t^L)$ denote the unmasking logits (corresponding to the output of policy network $f_\phi$, same as in Section 3.2). Interpreting these scores as unnormalized utilities associated with choosing each token for unmasking, under the PL model the likelihood of any particular *permutation* $\boldsymbol{\sigma} := (\sigma_1, \ldots, \sigma_L)$ of the tokens is given by

$$P(\boldsymbol{\sigma} \mid \boldsymbol{b}_t) = \prod_{l \in [L]} \frac{\exp(b_t^{\sigma_l})}{\sum_{j \geq l}^{L} \exp(b_t^{\sigma_j})} \ .$$

Informally, this corresponds to sampling all indices without replacement, where at each step the probability of choosing an item is proportional to its (exponentiated) utility.

The Plackett-Luce model can be easily adapted to model sampling of a fixed-length *ordered* set $\mathcal{U}_t \subseteq [L]$ where $|\mathcal{U}_t| = K$. This is because the probability of any particular partial permutation is equal to the marginalization over all permutations which complete it. Conretely, let $\Sigma(\mathcal{U}_t)$ be the set of permutations which begin with the sequence $\mathcal{U}_t$, then it follows that

$$P(\mathcal{U}_t \mid \boldsymbol{b}_t) = \sum_{\boldsymbol{\sigma} \in \Sigma(\mathcal{U}_t)} \underbrace{\prod_{l \in [L]} \frac{\exp(b_t^{\sigma_l})}{\sum_{j \geq l}^{L} \exp(b_t^{\sigma_j})}}_{P(\boldsymbol{\sigma} \mid \boldsymbol{b}_t)} = \prod_{l \in \mathcal{U}_t} \frac{\exp(b_t^{\sigma_i})}{\sum_{j \in \mathcal{U}_t^C \cup \mathcal{U}_t^{\geq l}} \exp(b_t^{\sigma_j})}$$

where $\mathcal{U}_t^C$ denotes the complement of $\mathcal{U}_t$, and $\mathcal{U}_t^{\geq l}$ denotes the indices which are in $\mathcal{U}_t$ at or after position $l$.

To extend the PL model to a *variable-length* unmasking sequence, we introduce a special *STOP* token with fixed utility $b_{\text{STOP}} = 0$, and proceed as follows:

1. Sample a single token $l \in [L]$ to unmask

$$l \sim \text{softmax}(\boldsymbol{b}_t / \tau_\pi)$$

   and initialize $\mathcal{U}_t^\pi = \{l\}$.

2. Sample another token $l'$ from the *renormalized distribution*

$$l' \sim \text{softmax}([\boldsymbol{b}_t^{\setminus \mathcal{U}}; b_{\text{STOP}}] / \tau_\pi)$$

   where $[\boldsymbol{b}_t^{\setminus \mathcal{U}}; b_{\text{STOP}}]$ denotes $\boldsymbol{b}_t$ concatenated with $b_{\text{STOP}}$ and with the logits of all previously selected indices $l \in \mathcal{U}_t^\pi$ set to $-\infty$.

3. Add $l'$ to $\mathcal{U}_t^\pi$.

4. If $l'$ was not *STOP*, repeat from 2.

As written above, this algorithm is not GPU-friendly due to the dynamic computation graph it implies. However, it can be implemented in an efficient manner by treating the loop in steps 2-4 as a Gumbel-argsort and simply masking out all actions which get sampled after *STOP*. Once the samples have been obtained, their likelihoods can be efficiently computed in the same manner as in the fixed-length case above by marginalizing over all actions which occur after *STOP*.

# G. Expert Steering

As mentioned in Section 4.2, we find that naïvely training confidence policies directly on full-diffusion generation ($BL = L$) task yields policies which beat out the heuristic methods, but underperform the ones trained in the semi-AR setting (cf. Figure 4).

Since semi-AR decoding lies within the function class representable by the policy $\pi_\phi$—that is, there is some $\phi$ such that $\pi_\phi$ approximates the semi-AR setting—we hypothesize that our failure to obtain similar policies is due to the vanishingly small probability of encountering autoregressive-like rollouts by chance. More concretely, imagine trying to train a policy on a task for which purely AR generation is substantially more effective than any other decoding strategy. In such a scenario, starting from a randomly initialized policy, the probability of observing a rollout resembling a purely AR sequence is approximately $1/L! \approx 0$ (where $L$ is the generation length), making it extremely unlikely to sample such rollouts during RL training.

To try to eschew this problem, we devise the *Expert Steering* (ES) training strategy as follows. Formally, letting $\pi_\phi$ denote the sampling policy to be learned, we simply replace it *at train time only* with the mixture

$$\pi_\phi^{ES}(\cdot \mid \boldsymbol{y}_t) = \frac{G}{G + E}\pi_\phi(\cdot \mid \boldsymbol{y}_t) + \frac{E}{G + E}\sum_{e=1}^{E}\delta_e(\cdot \mid \boldsymbol{y}_t)$$

where $G$ is the GRPO group size, $E$ is the number of experts to mimic, and each Dirac distribution $\delta_e$ represents a chosen deterministic "expert policy" (e.g., a heuristic method). Then in the outer loop of GRPO, instead of sampling a group $\boldsymbol{g} \sim \pi_\phi$ of size $G$ from $\pi_\phi$, we simply sample an augmented group $\boldsymbol{g}' \sim \pi_\phi^{ES}$ of size $G + E$. In the inner loop of GRPO we proceed as normal, making sure to use $\pi_\phi^{ES}$ when calculating the likelihood ratios to avoid the training instabilities that would otherwise arise (as samples from the Diracs may have likelihood $\approx 0$ under $\pi_\phi$)

In practice, for our experiments in Section 4.2 we use a single deterministic expert $\delta_e$ corresponding to Fast-dLLM with $\lambda = 0.9$ and $BL = 32$, and force exactly one draw ($E = 1$) from this heuristic per group. The effect is that if the policy is *worse* than this heuristic, that single sample will tend to have positive advantage, causing the policy to be biased toward it. On the other hand, if the policy is *better* than the heuristic, it will have negative advantage, causing it to be made *less* likely. Thus, expert steering encourages exploration towards the expert—in this case, Fast-dLLM—while still allowing the policy to go beyond it, thanks to the group relative loss in GRPO.

# H. Extended Background

**Masked vs. uniform discrete diffusion.** The D3PM framework (Austin et al., 2021a) characterizes forward corruption via a Markov kernel $Q_t$. Two canonical choices are (i) *masked* (absorbing-state) diffusion, which maps every token toward a distinguished mask state $M$,

$$Q_t^{\mathrm{mask}} \;=\; (1 - \alpha_t)I \;+\; \alpha_t \, \mathbf{1}e_m^\top,$$

with $\alpha_t \in [0, 1]$, $e_m$ denotes a one-hot vector (with dimension $V$) for $M$ token, and $\mathbf{1}$ represents a vector of all 1s of size $V$, and (ii) *uniform-state* diffusion, which replaces tokens uniformly at random,

$$Q_t^{\mathrm{uni}} \;=\; (1 - \alpha_t)I \;+\; \frac{\alpha_t}{V} \, \mathbf{1}\mathbf{1}^\top.$$

These differ in their limiting behavior and reverse constraints: $Q_t^{\mathrm{mask}}$ has a point-mass stationary distribution at $M$ and induces a monotonic forward trajectory in the mask count (reverse steps deterministically move from $M \to$ token), whereas $Q_t^{\mathrm{uni}}$ has a uniform stationary distribution and permits token $\leftrightarrow$ token substitutions in both directions. The absorbing choice recovers the familiar MDM objective (cf. Equation (1)) under a particular reweighting (Ou et al., 2025) and thus ties masked LMs to discrete diffusion (Austin et al., 2021a).

**Continuous-time discrete diffusion.** Recent work derives continuous-time limits (CTMC) for discrete diffusion with absorbing corruption and shows that the training objective is a *weighted time integral of token-wise cross-entropy*, with weight proportional to a signal-to-noise ratio term (e.g., $\alpha_t/(1 - \alpha_t)$). This yields a simple, schedule-invariant objective, clarifies forward–reverse consistency, and improves optimization and sampling without changing the prediction target (Shi et al., 2024; Sahoo et al., 2024). Conceptually, this places masked diffusion on the same ODE/SDE footing as continuous-state diffusion (Kingma & Gao, 2023) while retaining native discrete token inference.

**Continuous-input diffusion for categorical data.** A complementary approach to discrete-state diffusion is to keep the diffusion process continuous in both time and input by operating on token embeddings. For example in (Dieleman et al., 2022), an orthogonal line embeds tokens into $\mathbb{R}^d$ and applies fully continuous diffusion in *both* time and input space. Training can be done with cross-entropy via score interpolation, and sampling uses ODE/SDE solvers with tools like classifier-free guidance. This preserves the continuous machinery but introduces an embedding decoding interface and relaxes exact discreteness during the trajectory.

**Summary of differences.**

- **Masked (absorbing)**: monotonic reverse dynamics ($M \to$ token only); objective reduces to weighted MDM; pairs naturally with unmask-only sampling policies used in this work.

- **Uniform-state**: non-monotone reverse dynamics (token $\leftrightarrow$ token); encourages broader exploration but typically requires remasking/substitution moves and would thus complicate policy design.

- **Continuous-input**: inherits continuous samplers and guidance; trades exact discreteness for an embedding path and decoding step.

We adopt the absorbing formulation in this work.

# I. Training and Policy Network Configuration

*Table 1.* Training and policy configuration for our main experiments.

| Category | Parameter | Value |
|---|---|---|
| Training | Learning rate | 3e-5 |
| | LR scheduler | Cosine |
| | Warmup steps | 100 |
| | Effective batch size | 16 |
| | Weight decay | 0.1 |
| | Max gradient norm | 0.2 |
| | Clipping factor ($\epsilon$) | 0.5 (0.2 for expert steering) |
| GRPO | Training data | GSM8k and MATH training set mixture |
| | Num train epochs | 1 |
| | Group size | 8 |
| | $\beta$ (KL penalty) | 0.0 |
| | Correctness reward $r(\boldsymbol{y}, \boldsymbol{y}_t)$ | We use d1 (Zhao et al., 2025b) rewards. |
| Policy Network | Number of transformer blocks | 1 |
| | Hidden dimension | 128 |
| | Feedforward dimension | 512 |
| | Number of heads | 2 |
| | Time embedding dim | 128 |
| | Total parameter count | 300K |

# J. Policy Architecture Diagram

$c$ = confidence     $m$ = mask     $t$ = timestep     $k$ = top-$k$     $d$ = hidden_dim

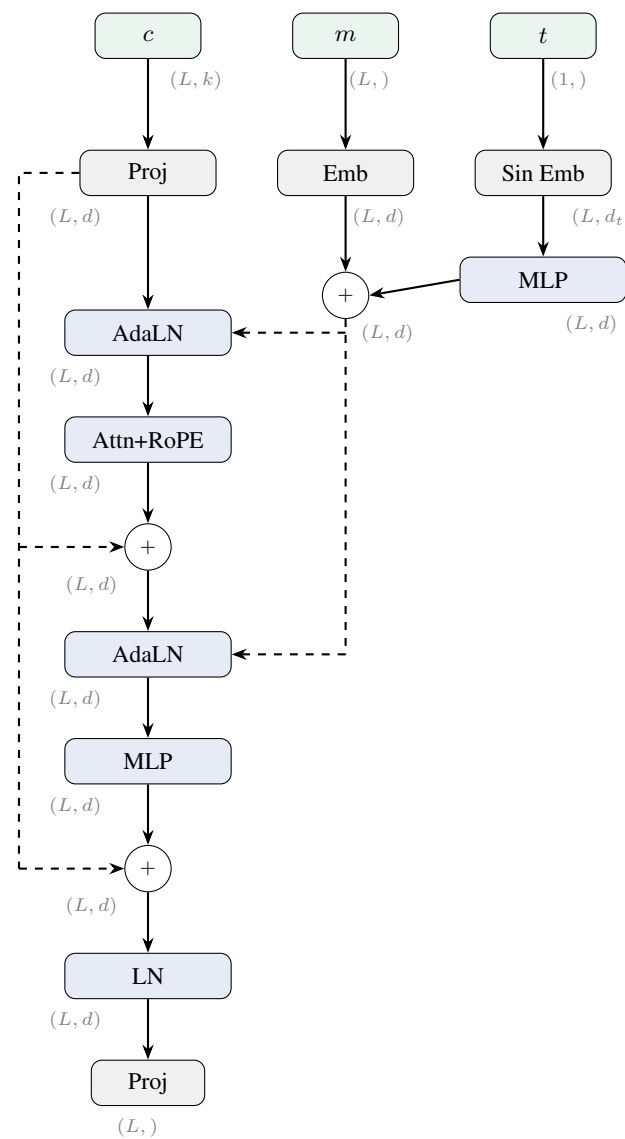

# K. Tabular Reference for Main LLaDA Experiments

In this appendix we include tabular results for the main LLaDA experiments so that future work can compare exact numbers more easily. All values correspond directly to the plotted points.



*Table 2.* $BL = 32$, GSM8K.

| Method | NFE | Accuracy (%) |
|---|---|---|
| Random | 8.0 | 20.6 |
| | 16.0 | $24.4 \pm 0.1$ |
| | 32.0 | $28.4 \pm 1.1$ |
| | 64.0 | $46.6 \pm 0.3$ |
| | 128.0 | $63.2 \pm 1.9$ |
| | 256.0 | $72.6 \pm 1.1$ |
| High Conf. | 8.0 | 20.6 |
| | 16.0 | 30.6 |
| | 32.0 | 46.9 |
| | 64.0 | 70.0 |
| | 128.0 | 76.1 |
| | 256.0 | 79.5 |
| Fast-dLLM | 10.6 | 17.5 |
| | 21.5 | 22.0 |
| | 28.7 | 39.7 |
| | 32.6 | 54.0 |
| | 37.1 | 63.0 |
| | 43.2 | 72.6 |
| | 51.9 | 76.6 |
| | 64.4 | 79.2 |
| | 82.2 | 79.7 |
| | 256.0 | 79.5 |
| Margin | 8.0 | 20.9 |
| | 16.0 | 30.6 |
| | 32.0 | 49.4 |
| | 64.0 | 69.6 |
| | 128.0 | 74.2 |
| | 256.0 | 78.8 |
| EB Sampler | 15.1 | 22.8 |
| | 23.0 | 29.2 |
| | 31.0 | 48.4 |
| | 37.6 | 64.6 |
| | 46.9 | 74.2 |
| | 54.9 | 77.6 |
| | 66.6 | 80.4 |
| | 249.0 | 80.8 |
| Ours | 10.3 | $35.2 \pm 0.3$ |
| | 46.2 | $75.5 \pm 0.7$ |
| | 50.7 | $76.6 \pm 0.2$ |
| | 82.7 | $80.0 \pm 0.3$ |
| | 135.8 | $80.5 \pm 0.5$ |

*Table 3.* $BL = 32$, MATH.

| Method | NFE | Accuracy (%) |
|---|---|---|
| Random | 8.0 | 7.6 |
| | 16.0 | $10.6 \pm 0.5$ |
| | 32.0 | $12.5 \pm 0.9$ |
| | 64.0 | $17.9 \pm 0.4$ |
| | 128.0 | $22.7 \pm 1.2$ |
| | 256.0 | $26.8 \pm 1.9$ |
| High Conf. | 8.0 | 7.6 |
| | 16.0 | 17.0 |
| | 32.0 | 20.4 |
| | 64.0 | 27.8 |
| | 128.0 | 33.2 |
| | 256.0 | 35.6 |
| Fast-dLLM | 11.4 | 7.4 |
| | 26.5 | 11.0 |
| | 36.9 | 16.6 |
| | 43.9 | 20.0 |
| | 49.7 | 26.4 |
| | 57.6 | 28.4 |
| | 68.1 | 32.4 |
| | 81.6 | 33.8 |
| | 100.7 | 34.4 |
| | 256.0 | 35.6 |
| Margin | 8.0 | 8.4 |
| | 16.0 | 14.2 |
| | 32.0 | 21.6 |
| | 64.0 | 27.8 |
| | 128.0 | 30.8 |
| | 256.0 | 35.8 |
| EB Sampler | 15.4 | 10.2 |
| | 23.8 | 16.0 |
| | 34.4 | 19.2 |
| | 44.8 | 25.6 |
| | 58.4 | 27.2 |
| | 69.0 | 33.8 |
| | 85.0 | 33.0 |
| | 251.0 | 36.0 |
| Ours | 10.8 | $15.3 \pm 0.2$ |
| | 59.1 | $31.5 \pm 1.0$ |
| | 64.6 | $31.9 \pm 0.6$ |
| | 100.2 | $34.1 \pm 1.0$ |
| | 153.4 | $35.1 \pm 0.1$ |



*Table 4.* $BL = 256$, GSM8K.

| Method | NFE | Accuracy (%) |
|---|---|---|
| Random | 8.0 | $19.5 \pm 0.5$ |
| | 16.0 | $30.6 \pm 0.8$ |
| | 32.0 | $39.8 \pm 0.9$ |
| | 64.0 | $47.3 \pm 0.1$ |
| | 128.0 | $48.9 \pm 1.1$ |
| | 256.0 | $51.6 \pm 1.2$ |
| High Conf. | 8.0 | 12.4 |
| | 16.0 | 37.3 |
| | 32.0 | 43.6 |
| | 64.0 | 38.9 |
| | 128.0 | 32.2 |
| | 256.0 | 33.6 |
| Fast-dLLM | 11.0 | 19.6 |
| | 15.8 | 32.2 |
| | 22.5 | 37.2 |
| | 32.1 | 38.7 |
| | 44.5 | 35.9 |
| | 60.8 | 33.8 |
| | 91.8 | 33.0 |
| | 256.0 | 33.6 |
| Margin | 8.0 | 20.1 |
| | 16.0 | 37.4 |
| | 32.0 | 45.9 |
| | 64.0 | 39.3 |
| | 128.0 | 30.4 |
| | 256.0 | 34.6 |
| EB Sampler | 10.5 | 11.6 |
| | 15.8 | 24.9 |
| | 21.8 | 32.6 |
| | 29.2 | 36.1 |
| | 43.6 | 29.5 |
| | 57.9 | 26.7 |
| | 74.2 | 35.0 |
| | 250.4 | 32.5 |
| Ours | 13.0 | $48.3 \pm 0.6$ |
| | 18.5 | $51.9 \pm 0.9$ |
| | 27.1 | $54.8 \pm 0.4$ |
| | 50.5 | $56.1 \pm 0.9$ |
| | 93.3 | $61.8 \pm 1.1$ |
| Ours (ES) | 15.7 | $50.6 \pm 0.8$ |
| | 20.6 | $51.9 \pm 0.2$ |
| | 70.5 | $77.2 \pm 0.4$ |
| | 72.4 | $76.7 \pm 0.5$ |
| | 87.9 | $75.1 \pm 0.1$ |

*Table 5.* $BL = 256$, MATH.

| Method | NFE | Accuracy (%) |
|---|---|---|
| Random | 8.0 | $11.5 \pm 1.0$ |
| | 16.0 | $13.5 \pm 0.2$ |
| | 32.0 | $14.8 \pm 0.8$ |
| | 64.0 | $17.9 \pm 1.0$ |
| | 128.0 | $18.3 \pm 0.2$ |
| | 256.0 | $18.5 \pm 2.7$ |
| High Conf. | 8.0 | 10.6 |
| | 16.0 | 16.6 |
| | 32.0 | 20.2 |
| | 64.0 | 20.8 |
| | 128.0 | 19.8 |
| | 256.0 | 20.2 |
| Fast-dLLM | 8.4 | 9.4 |
| | 17.4 | 12.6 |
| | 26.6 | 16.4 |
| | 37.2 | 17.4 |
| | 49.9 | 20.8 |
| | 64.5 | 18.4 |
| | 81.9 | 19.6 |
| | 106.7 | 20.2 |
| | 256.0 | 20.2 |
| Margin | 8.0 | 12.4 |
| | 16.0 | 18.4 |
| | 32.0 | 20.2 |
| | 64.0 | 20.4 |
| | 128.0 | 17.6 |
| | 256.0 | 21.6 |
| EB Sampler | 12.0 | 10.0 |
| | 19.2 | 13.4 |
| | 26.5 | 16.8 |
| | 35.4 | 17.8 |
| | 52.0 | 18.4 |
| | 69.0 | 17.6 |
| | 87.6 | 21.4 |
| | 250.8 | 20.2 |
| Ours | 14.5 | $18.8 \pm 1.8$ |
| | 22.3 | $19.8 \pm 1.0$ |
| | 33.7 | $19.9 \pm 0.4$ |
| | 61.6 | $21.0 \pm 1.3$ |
| | 121.5 | $22.9 \pm 1.0$ |
| Ours (ES) | 17.5 | $17.7 \pm 1.1$ |
| | 23.1 | $19.8 \pm 0.2$ |
| | 86.7 | $31.4 \pm 0.5$ |
| | 90.1 | $30.1 \pm 1.8$ |
| | 98.0 | $31.0 \pm 0.8$ |

