# OpenReview forum: "Learning Unmasking Policies for Diffusion Language Models"
_ICML.cc/2026/Conference — ICML 2026 spotlight_

### Official Review · Reviewer_bxr8 · 2026-02-19

**Soundness:** 4
**Presentation:** 4
**Significance:** 2
**Originality:** 3
**Overall Recommendation:** 5
**Confidence:** 4

**Summary:**

This paper proposes a reinforcement learning (RL) framework for learning a policy that determines the order of unmasking for diffusion language models (dLMs), in contrast to prior works that rely on heuristic rules (e.g. token confidence above some threshold). Unlike previous applications of RL to dLMs, this work does not aim to further train the underlying dLM for better reasoning, but instead trains a lightweight policy network that predicts unmasking positions to produce correct responses while minimizing computational budget. On math benchmarks, the authors find comparable performance of their method and heuristic-based approaches for semi-autoregressive regimes and improved performance in the full-sequence diffusion regimes.

**Compliance With Llm Reviewing Policy:**

Affirmed.

**Final Justification:**

During the rebuttal period the authors addressed my questions and concerns. I have raised my score to 5 - Accept

**Key Questions For Authors:**

- The input structure for the policy network seems perhaps ‘redundant’. For example, many MDM works drop $t$ from the network input because it can be inferred from the masking pattern $\mathbf{m}_t$. Moreover, for already decoded tokens, presumably some sentinel value is used in $\mathbf{c}_t$ (please correct me if I am wrong. As an aside, it would be useful to clarify / make explicit how these already decoded positions are handled in $\mathbf{c}_t$), therefore in theory $\mathbf{m}_t$ could be inferred from $\mathbf{c}_t$? Could the authors elaborate on whether they found each input to be essential?
- In the paragraph of Section 3.2 (lines 153-156, RHS) it was not clear to me why “avoiding constructing policies that operate at the token level” is a desiderata? Why is this approach not a good one?
- Could the authors clarify the role of the $1 / (T - \hat{T}_g)$ normalization term in the objective on page 4? Typically, these sequences are normalized by the generated sequence length, right? Why did the authors use this normalization instead?
- I found the last paragraph on the RHS column of page 5 to be a bit unintuitive. Why would the policy need to “slow down” in the last / later blocks? One might reason that in a sense these blocks are ‘easier’ because they carry more relative context compared to earlier blocks.
- Why do the results for “High Confidence" and fast-dLLM degrade in the sequence length generalization plot (Figure 6d)? I don’t see anything about these methods that would lead to this drop in performance. Is decoding for this plot done semi-AR or full-sequence?
- Is there potentially an issue with Figure 28? Why does the sequence consist of only EOT tokens with positions more to the ‘left’ decoded ‘late’?

**Limitations:**

Yes

**Strengths And Weaknesses:**

### **Strengths**
- The paper is well written and easy to follow.
- The implementation of the lightweight policy network is intuitive and appealing from a practical perspective.
- The ablations in the experimental section are extensive and contribute positively to the understanding of the method.
- The authors openly and honestly discuss the limitations of their method, e.g., the non-smooth sensitivity to the $\alpha$ parameter.

### **Weaknesses**
- The main weakness in my opinion is one that the authors themselves highlight. Namely, their trade-off of performance and compute is quite sensitive to the $\alpha$ parameter without an apparent “kink” in the curve or a smooth relationship to impact on NFEs. In essence, while no longer heuristic-based, this work seems to have replaced one hyperparameter (e.g., confidence threshold $\lambda$) with another, $\alpha$. Moreover, the need to re-train for each $\alpha$ is a significant drawback relative to methods, such as fast-dLLM, which can seamlessly navigate the pareto frontier at test time with no additional training or modifications.
    - Perhaps this is an issue of capacity. Did the authors explore making the policy network more expressive via additional parameters? I know there is a discussion where changing input type to hidden state requires more parameters which does not lead to improved performance. But what if the $\mathbf{c}_t$ input type is kept the same but parameters are increased. Even if the authors scale the network to 10x or 100x the size, if the results show significant improvement, I think this would be appealing to practitioners since this is still << the size of the underlying dLM.
- Additionally, the main benefits seem to come in the “full sequence” diffusion regime, however here we see a significant degradation in performance relative to semi-AR decoding. The only way to mitigate this degradation is through the expert steering solution, which seems like a ‘patch’ that masks a more fundamental limitation of the policy network’s ability to explore decoding orders, as the authors themselves note.
   - Perhaps the authors can explore a different setting where the left-to-right bias is less pronounced and therefore full sequence diffusion would be of more use?
- The authors could make their baselines more robust by including other decoding schemes, e.g. confidence-margin (Kim et al., 2025) and the EB sampler (Beh-Hamu et al., 2025).

**Minor weaknesses / suggestions**

- The switching of notation for sequence length dimension between $d$ and $L$ could be standardized / cleaned up.
- Relatedly, why did the authors switch from using $\mathbf{x}$ to denote sequences during training to $\mathbf{y}$ to denote sequences during generation?
- In line 57 RHS column, I would suggest citing Shi et al 2024 and Sahoo et al 2024 in addition to Ou et al 2025.
- In line 93 RHS column, I would recommend adding Kim et al, 2025 to the citations.
- In line 120 RHS column, perhaps it would be beneficial for readability to reiterate what $\tau$ is and how it interacts with the transition dynamics $P$.

---

**References**

Ben-Hamu, Heli, et al. "Accelerated sampling from masked diffusion models via entropy bounded unmasking." arXiv preprint arXiv:2505.24857 (2025).

Kim, Jaeyeon, et al. "Train for the worst, plan for the best: Understanding token ordering in masked diffusions." arXiv preprint arXiv:2502.06768 (2025).

Sahoo, Subham, et al. "Simple and effective masked diffusion language models." Advances in Neural Information Processing Systems 37 (2024): 130136-130184.

Shi, Jiaxin, et al. "Simplified and generalized masked diffusion for discrete data." Advances in neural information processing systems 37 (2024): 103131-103167.

---

> ### Author Rebuttal · Authors · 2026-03-30
>
> We thank you for your detailed feedback and many interesting questions.
>
> ✅ **Please find rebuttal figures [HERE](https://www.notion.so/Rebuttal-Figures-331e50384943800a91dcf50688fbe47f).**
>
> > *the need to re-train for each is a significant drawback…*
>
> **During rebuttal we ran a new experiment scaling the Bernoulli probabilities at test time via a factor $\beta$. As shown in Figure R2, $\beta$ provides a simple test-time knob, enabling effective traversal of the Pareto frontier without retraining.** Encouragingly, our test-time pareto even outperforms Fast-dLLM in low-NFEs regime under semi-AR, for example on MATH test-time scaling of policy achieves ~20% accuracy versus ~10% for Fast-dLLM at ~25 NFEs. We will include this in the camera-ready.
>
> > *…Did the authors explore making the policy network more expressive via additional parameters?*
>
> In the early stages of the project, we did experiment with larger policies (still using confidence inputs); however, we did not observe any significant improvements. Hence, we settled on the proposed lightweight design, as we aimed to keep the overhead to a minimum. That said, we agree that further scaling of the policy network, alongside the data used to train it, represents a promising direction for future work.
>
> > *…different setting where the left-to-right bias is less pronounced and therefore full sequence diffusion would be of more use?*
>
> We share your intuition and find application of our approach beyond text a very exciting direction. Since our policy operates solely on confidences, it is modality-agnostic and should be directly applicable to multi-modal MDMs. We highlight this as one of the most promising avenues for future work in Section 5, but feel that for the present work this is a bit out of scope.
>
> > *...make their baselines more robust…*
>
> **During the rebuttal, we ran the two suggested baselines (see Figure R4). We observe similar findings: our policies match them in the semi-AR regime while outperforming them in the full-diffusion.** We will add these in the camera-ready.
>
> > *The input structure for the policy network seems perhaps ‘redundant’...*
>
> **For rebuttal, we ran test-time ablations where, during generation, we set to zero either (i) time, (ii) masks, or (iii) both. As seen in Figure R3, the policy appears to rely on both inputs, as removing them leads to a 3–8% drop in performance.** We will add these to the camera-ready. Regarding the sentinel value: we do not use any and instead rely on dLLM’s confidence for unmasked positions as well. We agree, however, that a simpler architecture relying only on confidences (with sentinel values) would be worth exploring in future work.
>
> > *why “avoiding constructing policies that operate at the token level” is a desiderata?*
>
> We should have been more precise that this is a desideratum when efficiency of sampling is taken into account. In the ablation in Figure 7d, where we experimented with a policy that takes hidden states as input, the policy size immediately increased to 300M because of the large hidden state dimension (4096 in LLaDA). When the efficiency is important, this can start to become problematic.
>
> > *... clarify the role of the $1/(T - \hat{T}_g)$ normalization*
>
> The term $1/(T - \hat{T}_g)$ normalizes the terminal reward by the number of MDP steps taken in the trajectory $g$. This distributes the sparse terminal reward evenly across the steps that contributed to it and is analogous to dividing by the sequence length in the AR setting, where each token requires its own forward pass (one step in the MDP).
>
> > *Why ... “slow down” in the last / later blocks?*
>
> In most cases, the numerical answer is generated in the last block. Concretely, for policy sampling with $\alpha = 10$, 85% of GSM8k test samples produce the numerical answer in the last block. The policy appears to pick up on this pattern and therefore exhibits slower generation in that stage, potentially to avoid syntactic or formatting errors.
>
> > *Why ... fast-dLLM degrade in the sequence length generalization plot (Figure 6d)?*
>
> For Figure 6d, we use the full-diffusion setting ($BL = L$). Therefore, the degradation of the confidence-based sampler arises for the same reason as in Figure 1, and we observe that the failure of confidence-based samplers without semi-AR crutch gets worse as $L$ increases.
>
> > *... issue with Figure 28?*
>
> Note that Fast-dLLM in the full-diffusion regime often starts by generating positions furthest from the prompt first (i.e., right-to-left; see Figure 5). This behavior can sometimes lead to degenerate answers such as the one in Figure 28. We included this figure as a qualitative illustration of the degraded performance of Fast-dLLM when not using semi-AR (see Figure 1).
>
> We thank you again for your valuable feedback - it helped make our paper stronger. We remain available for any follow-up questions.

---

> > ### Author Rebuttal · Reviewer_bxr8 · 2026-04-01
> >
> > Thank you for this extensive response.
> >
> > I appreciate the new test-time $\beta$ experiment and believe it alleviates my concern about needing to re-train.
> >
> > I am somewhat surprised that larger auxiliary networks did not yield additional benefits in the authors initial experiments. This raises some doubts in my mind about the efficacy of the method
> >
> > I still maintain that focusing on benchmarks where semiAR decoding is beneficial made it hard to sell the method. That is, we don't see any improvement really for the semi-AR regime, which is how these dLLMs are used in practice, and so it makes it less compelling that gains were only observed for full-sequence decoding. I understand though that the rebuttal period is limited and therefore it is difficult to stand-up and run an experiment that would depart from the left-to-right regime. As a final suggestion, previous works have used Sudoku as interesting test-bed for showing the benefits of flexible decoding. Perhaps this is a useful dataset here as well, since it is small-scale and base models for these Sudoko experiments have relatively few parameters.
> >
> > Finally, my questions were answered. Thank you.

---

> > > ### Author Response · Authors · 2026-04-05
> > >
> > > We are delighted to hear that we have answered your questions, and that the new test-time $\beta$ experiment alleviated your concerns about the need to re-train.
> > >
> > > > I am ... surprised that larger ... networks did not yield additional benefits…
> > >
> > > While we agree that this is at first surprising, we believe it becomes less so when taking into account that our experiments train the policy *from scratch*. GRPO has primarily been validated in the context of LLMs that have already gone through both pre- and mid-training, putting them in a part of the weight space that may allow for more stable RL dynamics. It is thus possible that larger policies would see gains when paired with pre-training of some form, such as behavior cloning of a heuristic. This is an exciting direction for future research that our paper could inspire.
> > >
> > > > ... As a final suggestion, previous works have used Sudoku as interesting test-bed for showing the benefits of flexible decoding
> > >
> > > We followed your suggestion and ran the Sudoku experiment using small-scale MDMs (following §6.2.2 of the EB-Sampler paper). We then trained a single policy with $\alpha = 1$ and varied $\beta$ to get a Pareto frontier.
> > >
> > > Results in Figure R5: the policy closely tracks the Fast-dLLM/EB-Sampler frontier; this is likely near-optimal given that those methods already preserve full accuracy in only ~6 NFEs. We also note that the accuracy/efficiency sweetspot (black dot) occurs at the default hyperparameters (α=1, β=1). This speaks to the efficacy of our method: simply specifying a suitable reward incorporating both correctness and compute, **without tuning either the reward weighting (α) or the test-time parameter (β), RL discovers the Pareto-optimal operating point on its own.**
> > >
> > > We were also able to run a small size ablation (all with confidence-based input), as requested in your initial review. Results in Figure R6: **both a smaller (30K) and larger (3M) variant slightly underperform our default (300K)**. During training, we observed that the smaller variant required more steps to converge. The larger model initially outperformed the default but suffered larger spikes during training and ultimately converged to a worse reward.
> > >
> > > We note that this setting is quite different from that in the paper, where we are targeting large diffusion LMs trained on general-purpose data and therefore avoided the use of bespoke or synthetic benchmarks. However, we are very grateful for your suggestion to use this setting as a more computationally feasible test-bed during the rebuttal period, and hope that you have found these additional results insightful.
> > >
> > > > … we don't see any improvement really for the semi-AR regime, which is how these dLLMs are used in practice, and so it makes it less compelling that gains were only observed for full-sequence decoding …
> > >
> > > We would like to politely push back against this sentiment by offering an alternative viewpoint.
> > >
> > > In our minds, **the full-sequence decoding gains are very compelling**: they suggest that using RL to learn policies may allow the community to **break free from the crutch of semi-AR generation**. Indeed, the fact that heuristic strategies need semi-AR to work well is arguably a big reason *why* this is currently “how these dLLMs are used in practice”.
> > >
> > > Concretely, note that the learned BL-256 policies obtain **>50% GSM8k accuracy in ~15 steps**, which is **twice as fast as the best-performing semi-AR baseline** (Figure 3). This is therefore not just a philosophical issue: semi-AR generation puts a hard ceiling on how fast sampling can proceed in dLLMs, and our work substantially improves the situation. While we do not fully close the gap in the high-NFE regime, suggesting that there is still more work to be done before this technique can be adopted by practitioners, we believe that our paper is nonetheless a clear step in the right direction.
> > >
> > > Furthermore, we **do see improvements even when constrained to the semi-AR** regime:
> > > - **Less performance degradation in the low-NFE range** than any baseline heuristic (even when only training with $\alpha = 1$ and using $\beta$ to increase speed; that is, without retraining; Figure R2)
> > > - **Better test-time scaling and sample diversity** (Figure R1)
> > > - **Qualitatively distinct and novel strategies**, such as carefully generating the final answer while allowing the preceding reasoning to be unmasked quickly (Figure 4 in the submitted manuscript, as well as Appendix D)
> > >
> > > In summary, we believe that both the semi-AR and full-sequence results we present in this paper are compelling and exciting.
> > >
> > > Thank you for your valuable feedback and suggestions throughout this discussion. Incorporating them has allowed us to strengthen several aspects of the paper, which we are very grateful for. We hope that this response has settled your remaining concerns. Should you still have any outstanding concerns, please do not hesitate to relay them to the AC, with whom communication is not constrained on our side.

---

### Official Review · Reviewer_SGnd · 2026-02-25

**Soundness:** 3
**Presentation:** 3
**Significance:** 3
**Originality:** 3
**Overall Recommendation:** 4
**Confidence:** 4

**Summary:**

This work proposes formalizing the dLLM decoding process as a Markov decision process. By treating the underlying dLLM as the 'environment', it trains a lightweight policy network to automatically determine the optimal unmasking order, thereby achieving an optimal trade-off between generation speed and accuracy.

**Compliance With Llm Reviewing Policy:**

Affirmed.

**Final Justification:**

Thanks to the author for the reply. I have improved my score.

**Key Questions For Authors:**

Please refer to the three points in the Weaknesses

**Limitations:**

yes

**Strengths And Weaknesses:**

## Strengths
1. Proposing a trained policy network to automatically determine the unmasking order, as opposed to relying on hand-crafted heuristic rules, is a highly compelling and innovative perspective.
2. The methodology is presented with high clarity, and the empirical evaluation is reasonably comprehensive.
3. The learned policy demonstrates strong generalization capabilities across different models, indicating high robustness.

## Weaknesses
1. The policy trained on mathematical tasks suffers a significant performance degradation when transferred to code generation tasks. This suggests that the network may merely be memorizing the dynamic confidence distributions specific to "mathematical reasoning," rather than learning fundamental unmasking rules applicable to general tasks.
2. The policy network relies solely on the single-token maximum confidence score output by the dLLM as its input. This implies that the policy is essentially performing "shallow pattern matching" over the base model's probability distributions, without actually capturing or learning the complex semantic dependencies between tokens.
3. During training for the full-diffusion setting, the method introduces "expert steering" derived from Fast-dLLM, thereby absorbing the "left-to-right" decoding bias inherent to semi-AR models. Consequently, its superiority over baselines (e.g., High Confidence and Fast-dLLM) in full-diffusion test scenarios might not result from discovering a genuinely superior decoding strategy, but rather from simply mimicking the semi-AR behavior. This hypothesis is further corroborated by the observation that the performance of various methods remains largely comparable in the semi-AR test setting.

---

> ### Author Rebuttal · Authors · 2026-03-30
>
> Many thanks for your constructive criticism.
>
> ✅ **Please find rebuttal figures [HERE](https://www.notion.so/Rebuttal-Figures-331e50384943800a91dcf50688fbe47f).**
>
> > *The policy trained on mathematical tasks suffers a significant performance degradation when transferred to code generation...*
>
> While it is true that our math-trained policies experience a performance drop when evaluated on coding (as we point out in Section 4.3), **we show in our experiments that the domain gap can largely be closed by retraining on coding data (see the yellow lines in Figures 6b/6c)**. This indicates that our proposed framework itself generalizes well—what varies across domains are the confidence patterns, not the method's ability to learn from them. Moreover, training policies on mixtures of domains is a very promising direction, which we outline as important future work in Section 5.
>
> Lastly, rather than viewing this as a limitation, which as shown in our work can be mostly overcome by expanding training data, we believe the presence of domain-specific confidence patterns is itself an interesting and important finding that could inform future research on designing improved samplers for dLLMs.
>
> > *... policy is essentially performing "shallow pattern matching" ... without actually capturing or learning the complex semantic dependencies...*
>
> We agree that our policies do not explicitly capture semantic dependencies between tokens (as they rely on confidence-based inputs), however, we respectfully disagree with the claim they merely perform *“shallow pattern matching.”* As shown in Figure 3, our policies match—and in the low-NFE regime even outperform—SOTA heuristics such as Fast-dLLM under semi-AR, while outperforming them substantially in the full-diffusion. Moreover, in Appendix D we show that the learned policies exhibit qualitatively very different unmasking behavior (e.g., they unmask adjacent tokens much less frequently; see Figure 20). **Taken together, we believe our quantitative and qualitative results provide strong evidence that our policies learn meaningful and effective sampling strategies, even when relying solely on confidence-based inputs.**
>
> We would also like to refer to our ablation in Section 4.4 (see *“Confidence-based policy input”* paragraph and Figure 7d), where we explored going beyond top-1 confidence inputs. Concretely, we experimented with (i) using the top-50 confidence scores and (ii) using the dLLM’s hidden states as inputs. However, in both cases we found that these alternatives did not improve performance compared to using only the top-1 confidence score.
>
> > *... its superiority over baselines (e.g., High Confidence and Fast-dLLM) in full-diffusion test scenarios might not result from discovering a genuinely superior decoding strategy, but rather from simply mimicking the semi-AR ...*
>
> While it is true that our Expert Steering (which uses semi-AR Fast-dLLM as an expert) appears to encourage a left-to-right generation pattern (see Figure 5), **the quantitative results clearly indicate that the learned full-diffusion policy is not simply mimicking expert's semi-AR behavior (see Figures 3a/3b)**. Concretely, Fast-dLLM with semi-AR decoding achieves  ~20% accuracy in ~25 NFEs on GSM8K, whereas our ES-policy with $\alpha=10$ under full-diffusion achieves  ~50% accuracy in ~16 NFEs. Thus, **the policy in full-diffusion is both more accurate and faster than its semi-AR expert.**
>
> > *... the performance of various methods remains largely comparable in the semi-AR test setting.*
>
> Here we would again like to refer to our qualitative analysis in Appendix D. Despite some semi-AR policies matching Fast-dLLM in terms of downstream performance, they exhibit noticeably different unmasking patterns. For instance, the policies appear to allocate computation more uniformly across blocks (see Figure 4) and are also less likely to unmask adjacent tokens (see Figure 20). We believe that the ability of RL to independently discover such novel sampling behaviors is an interesting finding.
>
> Lastly, in response to other reviews (`4U4j`, `bxr8`) **we conducted [new experiments](https://www.notion.so/Rebuttal-Figures-331e50384943800a91dcf50688fbe47f) showing encouraging results for policy sampling under semi-AR**: (i) effective test-time Pareto frontiers via scaling of unmasking probabilities, with particularly strong results in the low-NFE regime (Figure R2), and (ii) improved performance under non-greedy decoding (Figure R1). Taken together, we believe these new results help us differentiate better from Fast-dLLM even in the semi-AR regime.
>
> We thank you again for reviewing our work. We remain available for any follow-up questions.

---

> > ### Author Rebuttal · Reviewer_SGnd · 2026-04-03
> >
> > Thanks to the author for the reply. I have improved my score.

---

> > > ### Author Response · Authors · 2026-04-04
> > >
> > > We are happy to hear that we have resolved all of your questions and concerns.
> > >
> > > Thank you for your valuable feedback throughout this discussion. Your concern about whether the policies truly found interesting and novel strategies made us reflect on how to improve the discussion of the results, both for expert steering and for the qualitative analysis. We are very grateful that you helped us articulate these findings more clearly, and will make sure that the final manuscript reflects these improvements.
> > >
> > > In general, your constructive engagement throughout this discussion has been very valuable to us. Thanks again for the time and care you put into reviewing our work.

---

### Official Review · Reviewer_sqCJ · 2026-03-12

**Soundness:** 2
**Presentation:** 3
**Significance:** 2
**Originality:** 2
**Overall Recommendation:** 3
**Confidence:** 3

**Summary:**

The paper proposed a training method to learning sampling procedures using RL, and formalized masked diffusion sampling as a Markov decision process. It also propose a lightweight policy model using one layer transformers to map the token confidences to unmasking decision. The experimental results show that the proposed method match the performance with SOTA heuristics methods.

**Compliance With Llm Reviewing Policy:**

Affirmed.

**Key Questions For Authors:**

1. What is the reason for not reporting results on popular benchmarks?
2. Why is a lightweight policy model used? What would happen if a larger one were used?
3. Why is there no comparison with other RL methods?

**Limitations:**

Yes

**Strengths And Weaknesses:**

## Strengths

1. The paper formalizes a Markov decision process and uses reinforcement learning to train this decision process.
2. The lightweight policy method is helpful for reinforcement learning.
3. The experimental results demonstrate that the proposed method matches the performance of state-of-the-art heuristic methods.

## Weaknesses

1. The advantages of the lightweight policy model are not discussed. Why not use a larger policy model?
2. It seems to be a direct application of GRPO to dLLMs, without a detailed discussion of related RL-based dLLM methods [1, 2, 3, 4].
3. There is a lack of comparison with similar methods [1, 2, 3, 4].
4. The paper does not report performance on popular benchmarks such as MMLU, GSM8K, etc.

[1] Zhao, Siyan, et al. "d1: Scaling reasoning in diffusion large language models via reinforcement learning." arXiv preprint arXiv:2504.12216 (2025).

[2] Yang, Jingyi, et al. "Taming masked diffusion language models via consistency trajectory reinforcement learning with fewer decoding step." arXiv preprint arXiv:2509.23924 (2025).

[3] He, Haoyu, et al. "Mdpo: Overcoming the training-inference divide of masked diffusion language models." arXiv preprint arXiv:2508.13148 (2025).

[4] Peng, Fred Zhangzhi, et al. "Path planning for masked diffusion model sampling." arXiv preprint arXiv:2502.03540 (2025).

---

> ### Author Rebuttal · Authors · 2026-03-30
>
> We thank you for your many questions.
>
> ✅ **Please find rebuttal figures [HERE](https://www.notion.so/Rebuttal-Figures-331e50384943800a91dcf50688fbe47f).**
>
> > *The advantages of the lightweight policy model are not discussed. Why not use a larger policy model?*
>
> > *Why is a lightweight policy model used? What would happen if a larger one were used?*
>
> **We did discuss the advantages of using a lightweight model in the manuscript—namely, that using a small unmasking policy introduces negligible computational overhead during sampling L.165.** Please also refer to Fig. 10, where we demonstrate that the overhead of our policy is minimal in terms of wall-clock time, thanks to its lightweight design.
>
> Following your comment, we will remind the reader that the unmasking policy must be invoked on top of the base dLLM at *every* sampling step: if a larger policy were to be used, the computational overhead would quickly accumulate. This could eliminate efficiency gains obtained from reducing the number of sampling steps. We would also like to highlight the ablation shown in Figure 7d, where we experimented with a larger policy (300M; hidden-state inputs). However, we found that it did not provide any advantage compared to our proposed lightweight design (300K; top-1 confidence inputs). We will add these nuances to L165.
>
> > *What is the reason for not reporting results on popular benchmarks?*
>
> > *...does not report performance on popular benchmarks such as MMLU, GSM8K, etc.*
>
> This seems to be an important misunderstanding. **We do report results on popular benchmarks—specifically GSM8K, MATH, HumanEval, and MBPP—which are the standard four datasets commonly used in the literature on sampling in MDMs [5, 6].** We are also somewhat confused by the remark that we do not evaluate on GSM8K, as it is one of the main datasets used throughout the paper (see, e.g., Fig. 1, Fig. 3, Fig. 6, and Fig. 7). Perhaps you had another benchmark in mind?
>
> Regarding MMLU, it is a multiple-choice dataset and is therefore commonly evaluated under a conditional likelihood regime. As such, it is typically not considered a generative task [7]. Our focus—similar to prior work on sampling in MDMs [5, 6]—is on tasks that require generating longer answers, which is why we did not include MMLU. Indeed, let us point out that none of the papers you referenced [1, 2, 3, 4] evaluate MMLU.
>
> > *It seems to be a direct application of GRPO to dLLMs, without a detailed discussion of related RL-based dLLM methods [1, 2, 3, 4].*
>
> > *There is a lack of comparison with similar methods [1, 2, 3, 4].*
>
> > *Why is there no comparison with other RL methods?*
>
> We would politely push back on the statement that our work is *“a direct application of GRPO to dLLMs.”* As we state in the introduction (L52, right column): *“…in contrast to recent works that use RL to improve the reasoning abilities of dLLMs [1], our goal is not to improve the capacity of the underlying model, but rather to use RL to automate the discovery of adaptive sampling strategies.”* In other words, **while most prior work uses RL to update the dLLM itself while keeping the sampling procedure fixed, we instead keep the dLLM frozen throughout GRPO and aim to *learn* the sampling procedure by introducing unmasking policies.** This distinction is most obviously reflected in the fact that our policy is a separate unmasking transformer, and not the dLLM itself (or its LoRA adaptation).  This is further reflected in our reward design (cf. Eq. 3), where in addition to the (verifiable) correctness signal, the reward also includes a computational term that encourages the discovery of faster sampling strategies.
>
> We would also like to point out that **we already cite d1 [1] and discuss how our approach differs from it in the introduction** (see the quote in the preceding paragraph). Regarding [2] and [3], similarly to [1], these works use RL to update the base dLLM itself, and we therefore view them as complementary to our effort of using RL to learn the sampling strategy instead. Hence, we believe [1, 2, 3] are not directly comparable to our work.
>
> Regarding [4], we acknowledge that we overlooked this paper. However, unlike our approach—where RL is used to learn adaptive samplers—in [4] the planner is trained in a supervised manner and the method employs a fixed unmasking schedule. Consequently, their focus is not on sampling efficiency (they use $T=L$ in their language experiments), which is one of the primary objectives of our work. We will add [4] to the related work in the camera-ready; we thank you for pointing it out.
>
> We hope these clarifications alleviate your concerns.
>
> [5] Wu et al. Fast-dLLM: Training-free Acceleration of Diffusion LLM by Enabling KV Cache and Parallel Decoding. ICLR 2026
>
> [6] Ben-Hamu et al. Accelerated Sampling from Masked Diffusion Models via Entropy Bounded Unmasking. NeurIPS 2025
>
> [7] Nie at al. Large Language Diffusion Models. NeurIPS 2025

---

> > ### Author Rebuttal · Reviewer_sqCJ · 2026-04-02
> >
> > Thank you for your reply. The rebuttal partially addressed my concerns. For benchmarks such as MMLU and GSM8K, the paper currently provides only figures showing the accuracies, but the exact values are difficult to read. I would appreciate it if the authors could include a table with the precise accuracy numbers for these benchmarks, along with direct comparisons to the baselines. Exact numerical results are more helpful than figures alone for evaluating performance.
> >
> > In addition, I would like to kindly remind the authors that the provided Notion link may reveal their identity and could potentially allow access to the rebuttal directory. I recommend using an anonymous GitHub repository to host the rebuttal results instead.

---

> > > ### Author Response · Authors · 2026-04-04
> > >
> > > We thank you for your continued engagement with our paper, and are happy to hear that we have managed to address some of your concerns.
> > >
> > > > For … MMLU and GSM8K, the paper currently provides only figures … I would appreciate it if the authors could include a table with the precise accuracy numbers …
> > >
> > > We should clarify that **we do not evaluate on MMLU** as this is typically treated as a likelihood-based (not generative) evaluation in this literature.
> > >
> > > That aside, we agree that **also showing** tabular results can make it easier to digest numerical results. **We have included tabular results below**, covering LLaDa on GSM8k for both BL=32 and BL=256 as requested by the reviewer. Note that these tables also include the additional *Margin* and *EB-Sampler* baselines that were added during the rebuttal.
> > >
> > > As our focus is on efficiency and presenting the entire Pareto frontiers of each method in an easy-to-digest format, we believe graphs are the best way of communicating our results in the main text. However, complementing those figures with tables is an excellent suggestion; we will therefore include tabular results for all the main experiments (Figure 3 and corresponding figures in the appendix) in a new appendix for the camera-ready version of the paper, so that readers can benefit from an alternative presentation of the results.
> > >
> > > Thank you for your valuable feedback and suggestions throughout this discussion. Incorporating them has allowed us to strengthen several aspects of the paper, which we are very grateful for. We hope that this response has settled your remaining concerns. Should you still have any outstanding concerns, please feel free to relay them to the AC, with whom communication is not constrained on our side.
> > >
> > > ---
> > >
> > > **LLaDA, GSM8k, BL=32**
> > >
> > > | Method | NFE | Accuracy (N/A = deterministic) |
> > > | --- | ---: | ---: |
> > > | Random | 8.0 | 20.55% ±0.00 |
> > > | Random | 16.0 | 24.44% ±0.11 |
> > > | Random | 32.0 | 28.35% ±1.10 |
> > > | Random | 64.0 | 46.60% ±0.30 |
> > > | Random | 128.0 | 63.23% ±1.93 |
> > > | Random | 256.0 | 72.61% ±1.06 |
> > > | High Confidence | 8.0 | 20.55% ±N/A |
> > > | High Confidence | 16.0 | 30.63% ±N/A |
> > > | High Confidence | 32.0 | 46.93% ±N/A |
> > > | High Confidence | 64.0 | 69.98% ±N/A |
> > > | High Confidence | 128.0 | 76.12% ±N/A |
> > > | High Confidence | 256.0 | 79.53% ±N/A |
> > > | Fast-dLLM | 10.6 | 17.51% ±N/A |
> > > | Fast-dLLM | 21.5 | 21.99% ±N/A |
> > > | Fast-dLLM | 28.7 | 39.65% ±N/A |
> > > | Fast-dLLM | 32.6 | 53.98% ±N/A |
> > > | Fast-dLLM | 37.1 | 63.00% ±N/A |
> > > | Fast-dLLM | 43.2 | 72.56% ±N/A |
> > > | Fast-dLLM | 51.9 | 76.57% ±N/A |
> > > | Fast-dLLM | 64.4 | 79.23% ±N/A |
> > > | Fast-dLLM | 82.2 | 79.68% ±N/A |
> > > | Fast-dLLM | 256.0 | 79.53% ±N/A |
> > > | Margin | 8.0 | 20.85% ±N/A |
> > > | Margin | 16.0 | 30.63% ±N/A |
> > > | Margin | 32.0 | 49.43% ±N/A |
> > > | Margin | 64.0 | 69.60% ±N/A |
> > > | Margin | 128.0 | 74.22% ±N/A |
> > > | Margin | 256.0 | 78.77% ±N/A |
> > > | EB Sampler | 15.1 | 22.82% ±N/A |
> > > | EB Sampler | 23.0 | 29.19% ±N/A |
> > > | EB Sampler | 31.0 | 48.37% ±N/A |
> > > | EB Sampler | 37.6 | 64.59% ±N/A |
> > > | EB Sampler | 46.9 | 74.22% ±N/A |
> > > | EB Sampler | 54.9 | 77.63% ±N/A |
> > > | EB Sampler | 66.6 | 80.36% ±N/A |
> > > | EB Sampler | 249.0 | 80.82% ±N/A |
> > > | Ours | 10.3 | 35.15% ±0.27 |
> > > | Ours | 46.2 | 75.51% ±0.68 |
> > > | Ours | 50.7 | 76.60% ±0.15 |
> > > | Ours | 82.7 | 80.04% ±0.30 |
> > > | Ours | 135.8 | 80.52% ±0.53 |
> > >
> > >
> > > **LLaDA, GSM8k, BL=256**
> > > | Method | NFE | Accuracy (N/A = deterministic) |
> > > | --- | ---: | ---: |
> > > | Random | 8.0 | 19.54% ±0.49 |
> > > | Random | 16.0 | 30.60% ±0.80 |
> > > | Random | 32.0 | 39.83% ±0.91 |
> > > | Random | 64.0 | 47.26% ±0.08 |
> > > | Random | 128.0 | 48.90% ±1.10 |
> > > | Random | 256.0 | 51.55% ±1.18 |
> > > | High Confidence | 8.0 | 12.43% ±N/A |
> > > | High Confidence | 16.0 | 37.30% ±N/A |
> > > | High Confidence | 32.0 | 43.59% ±N/A |
> > > | High Confidence | 64.0 | 38.89% ±N/A |
> > > | High Confidence | 128.0 | 32.15% ±N/A |
> > > | High Confidence | 256.0 | 33.59% ±N/A |
> > > | Fast-dLLM | 11.0 | 19.56% ±N/A |
> > > | Fast-dLLM | 15.8 | 32.15% ±N/A |
> > > | Fast-dLLM | 22.5 | 37.23% ±N/A |
> > > | Fast-dLLM | 32.1 | 38.74% ±N/A |
> > > | Fast-dLLM | 44.5 | 35.94% ±N/A |
> > > | Fast-dLLM | 60.8 | 33.81% ±N/A |
> > > | Fast-dLLM | 91.8 | 32.98% ±N/A |
> > > | Fast-dLLM | 256.0 | 33.59% ±N/A |
> > > | Margin | 8.0 | 20.09% ±N/A |
> > > | Margin | 16.0 | 37.38% ±N/A |
> > > | Margin | 32.0 | 45.87% ±N/A |
> > > | Margin | 64.0 | 39.27% ±N/A |
> > > | Margin | 128.0 | 30.40% ±N/A |
> > > | Margin | 256.0 | 34.57% ±N/A |
> > > | EB Sampler | 10.5 | 11.60% ±N/A |
> > > | EB Sampler | 15.8 | 24.94% ±N/A |
> > > | EB Sampler | 21.8 | 32.60% ±N/A |
> > > | EB Sampler | 29.2 | 36.09% ±N/A |
> > > | EB Sampler | 43.6 | 29.49% ±N/A |
> > > | EB Sampler | 57.9 | 26.69% ±N/A |
> > > | EB Sampler | 74.2 | 35.03% ±N/A |
> > > | EB Sampler | 250.4 | 32.45% ±N/A |
> > > | Ours | 13.0 | 48.29% ±0.61 |
> > > | Ours | 18.5 | 51.91% ±0.87 |
> > > | Ours | 27.1 | 54.81% ±0.42 |
> > > | Ours | 50.5 | 56.05% ±0.91 |
> > > | Ours | 93.3 | 61.84% ±1.14 |
> > > | Ours (with ES) | 15.7 | 50.57% ±0.76 |
> > > | Ours (with ES) | 20.6 | 51.91% ±0.19 |
> > > | Ours (with ES) | 70.5 | 77.15% ±0.42 |
> > > | Ours (with ES) | 72.4 | 76.65% ±0.53 |
> > > | Ours (with ES) | 87.9 | 75.06% ±0.08 |

---

### Official Review · Reviewer_4U4j · 2026-03-12

**Soundness:** 4
**Presentation:** 3
**Significance:** 4
**Originality:** 4
**Overall Recommendation:** 5
**Confidence:** 4

**Summary:**

This paper tackles the problem of designing better sampling strategies for masked diffusion language models. Rather than relying on hand-crafted heuristics like confidence thresholding, the authors frame the token unmasking process as a Markov Decision Process and train a lightweight policy network via GRPO to decide which tokens to unmask at each diffusion step. The policy takes token confidences as input and outputs per-position Bernoulli unmasking decisions. Training uses a multiplicative reward that balances correctness with computational efficiency The paper evaluates on GSM8k and MATH-500 using LLaDA-8B and Dream-7B, showing that learned policies match Fast-dLLM performance in the semi-autoregressive regime and clearly outperform all heuristics in the full-diffusion setting. Transferability across models, domains, and sequence lengths is also studied, alongside extensive qualitative analysis of unmasking behaviors.

**Compliance With Llm Reviewing Policy:**

Affirmed.

**Key Questions For Authors:**

1. How many GPU hours does the full RL training pipeline require (per α, per seed), and how does this compare to the effort of tuning λ for Fast-dLLM via grid search on a validation set?

2. For the expert steering experiments, how sensitive are the results to the choice of expert policy? Have you tried using random sampling or high-confidence unmasking as the expert instead of Fast-dLLM?

3. The paper uses greedy decoding (τ=0) throughout. Have you investigated whether learned policies also help when τ>0? This matters for applications requiring diverse samples (e.g., best-of-N or majority voting).

4. The policies are trained from scratch without KL regularization. Did you observe any mode collapse or degenerate behaviors during training beyond the instability at α=10?

**Limitations:**

1. The approach inherits the base dLLM's generation quality, it cannot improve upon what the frozen model is capable of producing, only change the order of unmasking.

2. The binary correctness reward limits applicability to tasks with easily verifiable answers. Extending to open-ended generation would require a fundamentally different reward design.

3. The fixed generation length L is pre-specified and not learned, meaning the policy cannot decide to generate shorter or longer sequences adaptively.

**Strengths And Weaknesses:**

### Strengths

1. Clean problem formulation. Casting dLLM sampling as an MDP is natural and well-motivated. The separation between the frozen dLLM (environment) and the lightweight policy (agent) is elegant, it avoids the heavy cost of RL post-training on the base model while still enabling adaptive sampling. The Bernoulli action formulation is simple, the closed-form likelihood is convenient for policy gradient methods, and the overall framework is easy to understand.

2. Strong empirical finding in the full-diffusion regime. The most compelling result is that learned policies substantially outperform heuristics when semi-AR constraints are removed. The results suggests that the semi-AR inductive bias may be a crutch that heuristics require but learned policies can partially overcome, which is a genuinely useful insight for the community.

3. Thorough qualitative analysis. Appendix D is very informative. The observation that the fast policy learns to slow down on the final numerical-answer block (Figure 4b), and that the expert-steered policy recovers left-to-right generation in the full-diffusion setting (Figure 5), provide real insight into what RL discovers.

4. Transferability experiments are encouraging. Model transfer working reasonably well speaks to the generality of confidence-based policies. The sequence-length generalizationis practically useful and nontrivial.

5. Honest discussion of limitations. The authors are commendably upfront about controllability issues with α, the failure of domain transfer to coding tasks, training instability at α=10, and the gap between full-diffusion and semi-AR policies.







### Weaknesses



1. The semi-AR result is arguably a negative finding for the main use case. In the BL=32 regime that most practitioners actually use, the learned policies essentially match but do not beat Fast-dLLM (Figures 3a/3c). The authors themselves speculate this may mean Fast-dLLM is near-optimal in this setting. Given the added complexity of RL training (hyperparameter search over α, training instability at α=10, seed sensitivity shown in Figure 13), the practical value proposition over simply tuning λ for Fast-dLLM is unclear for the standard operating regime.

2. Controllability. The paper acknowledges but does not resolve the issue that varying α does not smoothly traverse the Pareto frontier. The non-monotonic behavior for α≥4 (Figure 12, Appendix C.4) is concerning, different α values collapse to either the α=3 or α=10 regime with nothing in between. In practice, users need predictable control over the speed-accuracy tradeoff, and training separate policies per α is expensive.

3. While the exploration motivation is sound, the ES approach of mixing in one Fast-dLLM sample per GRPO group is handcrafted. The training instability it introduces (Section 4.2) is a practical concern. There is no systematic study of how the choice of expert, the number of expert samples E, or the expert's hyperparameters (λ=0.9, BL=32) affect outcomes.

---

> ### Author Rebuttal · Authors · 2026-03-30
>
> We thank you for your detailed feedback.
>
> ✅ **Please find rebuttal figures [HERE](https://www.notion.so/Rebuttal-Figures-331e50384943800a91dcf50688fbe47f).**
>
> > *The semi-AR result is arguably a negative finding*
>
> We agree that matching (rather than exceeding) Fast-dLLM in the semi-AR setting is perhaps our weakest result. However, these policies exhibit noticeably different unmasking patterns (see Appendix D): more uniform compute allocation across blocks (Figure 4) and less frequent unmasking of adjacent tokens (Figure 20). Hence, we find RL's ability to discover such novel sampling behaviors to be interesting and potentially of value to future research on sampling in dLLMs.
>
> > *Controllability…users need predictable control over the speed-accuracy tradeoff…*
>
> For the rebuttal **we conducted a new experiment scaling the Bernoulli probabilities at test time via a factor $\beta$. As shown in Figure R2, $\beta$ provides a simple test-time knob, enabling effective traversal of the Pareto frontier without retraining.** Encouragingly, our test-time pareto even outperforms Fast-dLLM in low-NFEs regime under semi-AR, for example on MATH, at around 25 NFEs,  test-time scaling of policy achieves ~20% accuracy versus ~10% for Fast-dLLM.
>
> >*…the ES approach of mixing in one Fast-dLLM sample per GRPO group is handcrafted…*
>
> > *…how sensitive are the results to the choice of expert policy?*
>
> Our goal with ES was simply to establish whether full-diffusion policies can be “nudged” towards more AR behavior without any form of SFT or behavioral cloning and we agree with you that further stabilizing training with ES is important future work (as we acknowledge in the paper). We focused on using Fast-dLLM as the expert since it is the strongest of the three semi-AR baselines considered. We did  briefly conduct preliminary experiments with multiple experts used in tandem, each an instance of Fast-dLLM with a different threshold, but did not observe any significant improvements compared to the single-expert case.
>
> > *How many GPU hours does the full RL training pipeline require…?*
>
> Training a single policy took, on average, about 24 hours on an 8×A100 GPU node. This is more expensive than tuning Fast-dLLM’s threshold, which can be done in a couple of hours. Improving training efficiency is therefore clearly an important direction for future work, perhaps by pretraining the policies or by reducing the effective batch size (which may be feasible, since the compute term in our reward helps preserve within-group reward diversity).
>
> > *…Have you investigated whether learned policies also help when τ>0?*
>
> During rebuttal, **we ran experiments comparing our policy with Fast-dLLM under non-greedy decoding (Figure R1). Policy sampling leads to improved test-time compute results** (majority voting and ORM), likely because the policy's stochastic Bernoulli sampling explores the solution space more effectively (as evident by better pass@k scaling). We will include this in the camera-ready and thank for your this suggestion, we believe this experiment will help us differentiate better from Fast-dLLM even in the semi-AR regime.
>
> > *…Did you observe any mode collapse or degenerate behaviors during training beyond the instability at α=10?*
>
> We did not observe instabilities beyond $\alpha=10$. The multiplicative reward structure was key to stable training (see Figures 7a/b). We did observe less stable training for larger policies (e.g., the hidden-state ablation in Figure 7d), possibly due to training from scratch without regularization.
>
> > *The approach inherits the base dLLM's generation quality…*
>
> Our focus was on learning fast samplers and not on improving the capability of the underlying dLLM (similar to how Fast-dLLM does not improve over LLaDA’s accuracy when unmasking one token at a time using “high-confidence”, see Figure 1). We agree, though, that efforts on learning fast samplers while simultaneously improving the capability of the underlying dLLM (e.g., via joint RL pre-training) represent a promising alternative direction.
>
> > *The binary correctness reward limits applicability…*
>
> Proposed multiplicative reward structure indeed requires binary rewards. However, we believe this can be relaxed without too many difficulties, for instance by retaining the additive reward but introducing the compute penalty progressively during training. In this way, the policy would initially focus on being correct and only later focus on being fast as well.
>
> > *The fixed generation length L is pre-specified…*
>
> This limitation is inherited from the base dLLMs, not specific to our approach. Since our policy uses a transformer, it can handle variable-length inputs without retraining (Figure 6d) and is compatible with future dLLMs that would adapt L “on the fly”.
>
> We thank you again for your valuable suggestions which made our paper stronger. We remain available for any follow-up questions.

---

> > ### Author Rebuttal · Reviewer_4U4j · 2026-04-02
> >
> > The Authors have fully solved all my questions. This is a technically solid paper which should be accepted.

---

> > > ### Author Response · Authors · 2026-04-04
> > >
> > > We are very happy to hear that we have resolved all of your questions.
> > >
> > > Thank you for your valuable feedback throughout this discussion. Incorporating your suggestion to evaluate under non-greedy decoding revealed encouraging results that really strengthened the paper's contribution in the semi-AR regime and helped us differentiate more clearly from the Fast-dLLM baseline, which we are very grateful for.
> > >
> > > In general, your detailed and constructive engagement throughout this discussion has been very valuable to us. Thanks again for the time and care you put into reviewing our work.

---

### Decision · Program_Chairs · 2026-04-30

**Decision:**

Accept (spotlight)

**Comment:**

The paper tackles the problem of sampling in diffusion language models. While most prior work relies on heuristic rules, the authors propose training a lightweight single-layer transformer policy network that guides the unmasking procedure. Results demonstrate improvements over existing heuristic methods, particularly in the full-diffusion sampling setting.

Reviewers rated soundness good-to-excellent, presentation good-to-excellent, significance good, and originality good, with recommendations of two Accepts, one Weak Accept, and one Weak Reject.

As main strength, reviewers mentioned the clean MDP formulation of sampling, strong empirical gains in the full-diffusion regime, and promising transferability experiments.

As key weakness, reviewers questioned the limited improvements in the practical semi-autoregressive regime.

During rebuttal, the authors provided new test-time scaling experiments to control the Pareto frontier without retraining, along with non-greedy decoding results showing better diversity and scaling. All reviewers acknowledged the rebuttal, with three fully resolved (and scores raised) and one partially resolved.

The paper is technically sound and well-written. It tackles an emerging problem in efficient sampling for diffusion language models and will be of high interest to the ICML community. I recommend Acceptance.